# APPROXIMABILITY OF DISCRIMINATORS IMPLIES DIVERSITY IN GANS

**Yu Bai**
Stanford University
yub@stanford.edu

**Tengyu Ma**
Stanford University
tengyuma@stanford.edu

**Andrej Risteski**
MIT
risteski@mit.edu

## ABSTRACT

While Generative Adversarial Networks (GANs) have empirically produced impressive results on learning complex real-world distributions, recent works have shown that they suffer from lack of diversity or mode collapse. The theoretical work of Arora et al. (2017a) suggests a dilemma about GANs' statistical properties: powerful discriminators cause overfitting, whereas weak discriminators cannot detect mode collapse.

By contrast, we show in this paper that GANs can in principle learn distributions in Wasserstein distance (or KL-divergence in many cases) with polynomial sample complexity, if the discriminator class has strong distinguishing power against the particular generator class (instead of against all possible generators). For various generator classes such as mixture of Gaussians, exponential families, and invertible and injective neural networks generators, we design corresponding discriminators (which are often neural nets of specific architectures) such that the Integral Probability Metric (IPM) induced by the discriminators can provably approximate the Wasserstein distance and/or KL-divergence. This implies that if the training is successful, then the learned distribution is close to the true distribution in Wasserstein distance or KL divergence, and thus cannot drop modes. Our preliminary experiments show that on synthetic datasets the test IPM is well correlated with KL divergence or the Wasserstein distance, indicating that the lack of diversity in GANs may be caused by the sub-optimality in optimization instead of statistical inefficiency.

## 1 INTRODUCTION

In the past few years, we have witnessed great empirical success of Generative Adversarial Networks (GANs) (Goodfellow et al., 2014) in generating high-quality samples in many domains. Various ideas have been proposed to further improve the quality of the learned distributions and the stability of the training. (See e.g., (Arjovsky et al., 2017; Odena et al., 2016; Huang et al., 2017; Radford et al., 2016; Tolstikhin et al., 2017; Salimans et al., 2016; Jiwoong Im et al., 2016; Durugkar et al., 2016; Xu et al., 2017) and the reference therein.)

However, understanding of GANs is still in its infancy. Do GANs actually learn the target distribution? Recent work (Arora et al., 2017a;b; Dumoulin et al., 2016) has both theoretically and empirically brought the concern to light that distributions learned by GANs suffer from mode collapse or lack of diversity — the learned distribution tends to miss a significant amount of modes of the target distribution (elaborated in Section 1.1). The main message of this paper is that the mode collapse can be in principle alleviated by designing proper discriminators with strong distinguishing power against specific families of generators such as special subclasses of neural network generators (see Section 1.2 and 1.3 for a detailed introduction.)

### 1.1 BACKGROUND ON MODE COLLAPSE IN GANS

We mostly focus on the Wasserstein GAN (WGAN) formulation (Arjovsky et al., 2017) in this paper. Define the $\mathcal{F}$-Integral Probability Metric ($\mathcal{F}$-IPM) (Müller, 1997) between distributions $p, q$ as

$$W_{\mathcal{F}}(p, q) := \sup_{f \in \mathcal{F}} \left| \mathbb{E}_{X \sim p}[f(X)] - \mathbb{E}_{X \sim q}[f(X)] \right|. \tag{1}$$

Given samples from distribution $p$, WGAN sets up a family of generators $\mathcal{G}$, a family of discriminators $\mathcal{F}$, and aims to learn the data distribution $p$ by solving

$$\min_{q \in \mathcal{G}} \ W_{\mathcal{F}}(\hat{p}^n, \hat{q}^m) \tag{2}$$

where $\hat{p}^n$ denotes "the empirical version of the distribution $p$", meaning the uniform distribution over a set of $n$ i.i.d samples from $p$ (and similarly $\hat{q}^m$.)

When $\mathcal{F} = \{$all 1-Lipschitz functions$\}$, IPM reduces to the Wasserstein-1 distance $W_1$. In practice, parametric families of functions $\mathcal{F}$ such as multi-layer neural networks are used for approximating Lipschitz functions, so that we can empirically optimize this objective eq. (2) via gradient-based algorithms as long as distributions in the family $\mathcal{G}$ have parameterized samplers. (See Section 2 for more details.)

One of the main theoretical and empirical concerns with GANs is the issue of "mode-collapse"(Arora et al., 2017a; Salimans et al., 2016) — the learned distribution $q$ tends to generate high-quality but low-diversity examples. Mathematically, the problem apparently arises from the fact that IPM is weaker than $W_1$, and the mode-dropped distribution can fool the former (Arora et al., 2017a): for a typical distribution $p$, there exists a distribution $q$ such that simultaneously the followings happen:

$$W_{\mathcal{F}}(p, q) \lesssim \varepsilon \text{ and } W_1(p, q) \gtrsim 1. \tag{3}$$

where $\lesssim, \gtrsim$ hide constant factors. In fact, setting $q = \hat{p}^N$ with $N = R(\mathcal{F})/\varepsilon^2$, where $R(\mathcal{F})$ is a complexity measure of $\mathcal{F}$ (such as Rademacher complexity), $q$ satisfies eq. (3) but is clearly a mode-dropped version of $p$ when $p$ has an exponential number of modes.

Reasoning that the problem is with the strength of the discriminator, a natural solution is to increase it to larger families such as all 1-Lipschitz functions. However, Arora et al. (Arora et al., 2017a) points out that Wasserstein-1 distance doesn't have good generalization properties: the empirical Wasserstein distance used in the optimization is very far from the population distance. Even for a spherical Gaussian distribution $p = \mathsf{N}(0, \frac{1}{d}I_{d \times d})$ (or many other typical distributions), when the distribution $q$ is exactly equal to $p$, letting $\hat{q}^m$ and $\hat{p}^n$ be two empirical versions of $q$ and $p$ with $m, n = \text{poly}(d)$, we have with high probability,

$$W_1(\hat{p}^n, \hat{q}^m) \gtrsim 1 \quad \text{even though} \quad W_1(p, q) = 0. \tag{4}$$

Therefore even when learning succeeds ($p = q$), it cannot be gleaned from the empirical version of $W_1$.

The observations above pose a dilemma in establishing the theories of GANs: powerful discriminators cause overfitting, whereas weak discriminators result in diversity issues because IPM doesn't approximate the Wasserstein distance The lack of diversity has also been observed empirically by (Srivastava et al., 2017; Di & Yu, 2017; Borji, 2018; Arora et al., 2017b).

## 1.2 An approach to diversity: discriminator families with restricted approximability

This paper proposes a resolution to the conundrum by designing a discriminator class $\mathcal{F}$ that is particularly strong against a specific generator class $\mathcal{G}$. We say that a discriminator class $\mathcal{F}$ (and its IPM $W_{\mathcal{F}}$) has *restricted approximability* w.r.t. a generator class $\mathcal{G}$ and the data distribution $p$, if $\mathcal{F}$ can distinguish $p$ and any $q \in \mathcal{G}$ approximately as well as all 1-Lipschitz functions can do:

$$W_{\mathcal{F}} \text{ has restricted approximability w.r.t. } \mathcal{G} \text{ and } p$$
$$\triangleq \ \forall q \in \mathcal{G}, \ \gamma_L(W_1(p, q)) \lesssim W_{\mathcal{F}}(p, q) \lesssim \gamma_U(W_1(p, q)), \tag{5}$$

where $\gamma_L(\cdot)$ and $\gamma_U(\cdot)$ are two monotone nonnegative functions with $\gamma_L(0) = \gamma_U(0) = 0$. The paper mostly focuses on $\gamma_L(t) = t^\alpha$ with $1 \leq \alpha \leq 2$ and $\gamma_U(t) = t$, although we use the term "restricted approximability" more generally for this type of result (without tying it to a concrete definition of $\gamma$). In other words, we are looking for discriminators $\mathcal{F}$ so that $\mathcal{F}$-IPM can approximate the Wasserstein distance $W_1$ for the data distribution $p$ and any $q \in \mathcal{G}$.

Throughout the rest of this paper, we will focus on the realizable case, that is, we assume $p \in \mathcal{G}$, in which case we say $W_{\mathcal{F}}$ *has restricted approximability with respect to* $\mathcal{G}$ if eq. (5) holds for all

$p, q \in \mathcal{G}$. We note, however, that such a framework allows the non-realizable case $p \notin \mathcal{G}$ in full generality (for example, results can be established through designing $\mathcal{F}$ that satisfies the requirement in Lemma 4.3).

A discriminator class $\mathcal{F}$ with restricted approximability resolves the dilemma in the following way.

First, $\mathcal{F}$ avoids mode collapse – if the IPM between $p$ and $q$ is small, then by the left hand side of eq. (5), $p$ and $q$ are also close in Wasserstein distance and therefore significant mode-dropping cannot happen. [1]

Second, we can pass from population-level guarantees to empirical-level guarantees – as shown in Arora et al. (2017a), classical capacity bounds such as the Rademacher complexity of $\mathcal{F}$ relate $W_{\mathcal{F}}(p, q)$ to $W_{\mathcal{F}}(\hat{p}^n, \hat{q}^m)$. Therefore, as long as the capacity is bounded, we can expand on eq. (5) to get a full picture of the statistical properties of Wasserstein GANs:

$$\forall q \in \mathcal{G}, \; \gamma_L(W_1(p, q)) \lesssim W_{\mathcal{F}}(p, q) \approx W_{\mathcal{F}}(\hat{p}^n, \hat{q}^m) \lesssim \gamma_U(W_1(p, q)).$$

Here the first inequality addresses the **diversity property** of the distance $W_{\mathcal{F}}$, and the second approximation addresses the **generalization** of the distance, and the third inequality provides the reverse guarantee that if the training fails to find a solution with small IPM, then indeed $p$ and $q$ are far away in Wasserstein distance.[2] To the best of our knowledge, this is the first theoretical framework that tackles the statistical theory of GANs with polynomial samples.

The main body of the paper will develop techniques for designing discriminator class $\mathcal{F}$ with restricted approximability for several examples of generator classes including simple classes like mixtures of Gaussians, exponential families, and more complicated classes like distributions generated by invertible neural networks. In the next subsection, we will show that properly chosen $\mathcal{F}$ provides diversity guarantees such as inequalities eq. (5).

## 1.3    DESIGN OF DISCRIMINATORS WITH RESTRICTED APPROXIMABILITY

We start with relatively simple families of distributions $\mathcal{G}$ such as Gaussian distributions and exponential families, where we can directly design $\mathcal{F}$ to distinguish pairs of distribution in $\mathcal{G}$. As we show in Section 3, for Gaussians it suffices to use one-layer neural networks with ReLU activations as discriminators, and for exponential families to use linear combinations of the sufficient statistics.

In Section 4, we study the family of distributions generated by invertible neural networks. We show that a special type of neural network discriminators with one additional layer than the generator has restricted approximability[3]. We show this discriminator class guarantees that $W_1(p, q)^2 \lesssim W_{\mathcal{F}}(p, q) \lesssim W_1(p, q)$ where here we hide polynomial dependencies on relevant parameters (Theorem 4.2). We remark that such networks can also produce an exponentially large number of modes due to the non-linearities, and our results imply that if $W_{\mathcal{F}}(p, q)$ is small, then most of these exponential modes will show up in the learned distribution $q$.

One limitation of the invertibility assumption is that it only produces distributions supported on the entire space. The distribution of natural images is often believed to reside approximately on a low-dimensional manifold. When the distribution $p$ have a Lebesgue measure-zero support, the KL-divergence (or the reverse KL-divergence) is infinity unless the support of the estimated distribution coincides with the support of $p$.[4] Therefore, while our proof makes crucial use of the KL-divergence in the invertible case, the KL-divergence is fundamentally not the proper measurement of the statistical distance for the cases where both $p$ and $q$ have low-dimensional supports.

The crux of the technical part of the paper is to establish the approximation of Waserstein distance by IPMs for generators with low-dimensional supports. We will show that a variant of an IPM can still be sandwiched by Wasserstein distance as in form of eq. (5) without relating to KL-divergence (Theorem 4.5). This demonstrates the advantage of GANs over MLE approach on learning distributions

---

[1] Informally, if most of the modes of $p$ are $\varepsilon$-far away from each other, then as long as $W_1(p, q) \ll \varepsilon$, $q$ has to contain most of the modes of $p$.

[2] We also note that the third inequality can hold for all $p, q$ as long as $\mathcal{F}$ is a subset of Lipschitz functions.

[3] This is consistent with the empirical finding that generators and discriminators with similar depths are often near-optimal choices of architectures.

[4] The formal mathematical statement is that $D_{\mathrm{kl}}(p \| q)$ is infinity unless $p$ is absolutely continuous with respect to $q$.

with low-dimensional supports. As the main proof technique, we develop tools for approximating the log-density of a smoothed neural network generator.

We demonstrate in synthetic and controlled experiments that the IPM correlates with the Wasserstein distance for low-dimensional distributions with measure-zero support and correlates with KL-divergence for the invertible generator family (where computation of KL is feasible) (Section 5 and Appendix G.) The theory suggests the possibility that when the KL-divergence or Wasserstein distance is not measurable in more complicated settings, the test IPM could serve as a candidate alternative for measuring the diversity and quality of the learned distribution. We also remark that on real datasets, often the optimizer is tuned to carefully balance the learning of generators and discriminators, and therefore the reported training loss is often not the test IPM (which requires optimizing the discriminator until optimality.) Anecdotally, the distributions learned by GANs can often be distinguished by a well-trained discriminator from the data distribution, which suggests that the IPM is not well-optimized (See Lopez-Paz & Oquab (2016) for analysis of for the original GANs formulation.) We conjecture that the lack of diversity in real experiments may be caused by sub-optimality of the optimization, rather than statistical inefficiency.

### 1.4 RELATED WORK

Various empirical proxy tests for diversity, memorization, and generalization have been developed, such as interpolation between images (Radford et al., 2016), semantic combination of images via arithmetic in latent space (Bojanowski et al., 2017), classification tests (Santurkar et al., 2017), etc. These results by and large indicate that while "memorization" is not an issue with most GANs, lack of diversity frequently is.

As discussed thoroughly in the introduction, Arora et al. (2017a;b) formalized the potential theoretical sources of mode collapse from a weak discriminator, and proposed a "birthday paradox" that convincingly demonstrates this phenomenon is real. Many architectures and algorithms have been proposed to remedy or ameliorate mode collapse (Dumoulin et al., 2016; Srivastava et al., 2017; Di & Yu, 2017; Borji, 2018; Lin et al., 2017) with varying success. Feizi et al. (2017) showed provable guarantees of training GANs with quadratic discriminators when the generators are Gaussians. However, to the best of our knowledge, there are no provable solutions to this problem in more substantial generality.

The inspiring work of Zhang et al. (Zhang et al., 2017) shows that the IPM is a proper metric (instead of a pseudo-metric) under a mild regularity condition. Moreover, it provides a KL-divergence bound with finite samples when the densities of the true and estimated distributions exist. Our Section 4.1 can be seen as an extension of (Zhang et al., 2017, Proposition 2.9 and Corollary 3.5). The strength in our work is that we develop statistical guarantees in Wasserstein distance for distributions such as injective neural network generators, where the data distribution resides on a low-dimensional manifold and thus does not have proper density.

Liang (2017) considers GANs in a non-parametric setup, one of the messages being that the sample complexity for learning GANs improves with the smoothness of the generator family. However, the rate they derive is non-parametric – exponential in the dimension – unless the Fourier spectrum of the target family decays extremely fast, which can potentially be unrealistic in practical instances.

The invertible generator structure was used in Flow-GAN (Grover et al., 2018), which observes that GAN training blows up the KL on real dataset. Our theoretical result and experiments show that successful GAN training (in terms of the IPM) does imply learning in KL-divergence when the data distribution can be generated by an invertible neural net. This suggests, along with the message in (Grover et al., 2018), that the real data cannot be generated by an invertible neural network. In addition, our theory implies that if the data can be generated by an *injective* neural network (Section 4.2), we can bound the closeness between the learned distribution and the true distribution in Wasserstein distance (even though in this case, the KL divergence is no longer an informative measure for closeness.)

## 2 PRELIMINARIES AND NOTATION

The notion of IPM (recall the definition in eq. (1)) includes a number of statistical distances such as TV (total variation) and Wasserstein-1 distance by taking $\mathcal{F}$ to be 1-bounded and 1-Lipschitz

functions respectively. When $\mathcal{F}$ is a class of neural networks, we refer to the $\mathcal{F}$-IPM as the *neural net IPM*.[5]

There are many distances of interest between distributions that are not IPMs, two of which we will particularly focus on: the KL divergence $D_{\mathrm{kl}}(p\|q) = \mathbb{E}_p[\log p(X) - \log q(X)]$ (when the densities exist), and the Wasserstein-2 distance, defined as $W_2(p,q)^2 = \inf_{\pi \in \Pi} \mathbb{E}_{(X,Y) \sim \pi}[\|X - Y\|^2]$ where $\Pi$ be the set of couplings of $(p,q)$. We will only consider distributions with finite second moments, so that $W_1$ and $W_2$ exist.

For any distribution $p$, we let $\hat{p}^n$ be the empirical distribution of $n$ i.i.d. samples from $p$. The Rademacher complexity of a function class $\mathcal{F}$ on a distribution $p$ is $R_n(\mathcal{F}, p) = \mathbb{E}\left[\sup_{f \in \mathcal{F}} |\frac{1}{n}\sum_{i=1}^n \varepsilon_i f(X_i)|\right]$ where $X_i \sim p$ i.i.d. and $\varepsilon_i \sim \{\pm 1\}$ are independent. We define $R_n(\mathcal{F}, \mathcal{G}) = \sup_{p \in \mathcal{G}} R_n(\mathcal{F}, p)$ to be the largest Rademacher complexity over $p \in \mathcal{G}$. The training IPM loss (over the entire dataset) for the Wasserstein GAN, assuming discriminator reaches optimality, is $\mathbb{E}_{\hat{q}^n}[W_{\mathcal{F}}(\hat{p}^n, \hat{q}^n)]$[6]. Generalization of the IPM is governed by the quantity $R_n(\mathcal{F}, \mathcal{G})$, as stated in the following result (see Appendix A.1 for the proof):

**Theorem 2.1** (Generalization, c.f. (Arora et al., 2017a)). *For any $p \in \mathcal{G}$, we have that*

$$\forall q \in \mathcal{G}, \ \mathbb{E}_{\hat{p}^n}|W_{\mathcal{F}}(p,q) - \mathbb{E}_{\hat{q}^n}[W_{\mathcal{F}}(\hat{p}^n, \hat{q}^n)]| \leq 4R_n(\mathcal{F}, \mathcal{G}).$$

**Miscellaneous notation.** We let $\mathsf{N}(\mu, \Sigma)$ denote a (multivariate) Gaussian distribution with mean $\mu$ and covariance $\Sigma$. For quantities $a, b > 0$ $a \lesssim b$ denotes that $a \leq Cb$ for a universal constant $C > 0$ unless otherwise stated explicitly.

# 3 RESTRICTED APPROXIMABILITY FOR BASIC DISTRIBUTIONS

## 3.1 GAUSSIAN DISTRIBUTIONS

As a warm-up, we design discriminators with restricted approximability for relatively simple parameterized distributions such Gaussian distributions, exponential families, and mixtures of Gaussians. We first prove that one-layer neural networks with ReLU activation are strong enough to distinguish Gaussian distributions with the restricted approximability guarantees.

We consider the set of Gaussian distributions with bounded mean and well-conditioned covariance $\mathcal{G} = \left\{ p_\theta = \mathsf{N}(\mu, \Sigma) : \|\mu\|_2 \leq D, \sigma_{\min}^2 I_d \preceq \Sigma \preceq \sigma_{\max}^2 I_d \right\}$. Here $D, \sigma_{\min}$ and $\sigma_{\max}$ are considered as given hyper-parameters. We will show that the IPM $W_{\mathcal{F}}$ induced by the following discriminators has restricted approximability w.r.t. $\mathcal{G}$:

$$\mathcal{F} := \left\{ x \mapsto \mathrm{ReLU}(v^\top x + b) : \|v\|_2 \leq 1, |b| \leq D \right\}, \tag{6}$$

**Theorem 3.1.** *The set of one-layer neural networks ($\mathcal{F}$ defined in eq. (6)) has restricted approximability w.r.t. the Gaussian distributions in $\mathcal{G}$ in the sense that for any $p, q \in \mathcal{G}$*

$$\kappa \cdot W_1(p,q) \lesssim W_{\mathcal{F}}(p,q) \leq W_1(p,q).$$

*with $\kappa = \frac{1}{\sqrt{d}} \frac{\sigma_{\min}}{\sigma_{\max}}$. Moreover, $\mathcal{F}$ satisfies Rademacher complexity bound $R_n(\mathcal{F}, \mathcal{G}) \lesssim \frac{D + \sigma_{\max}\sqrt{d}}{\sqrt{n}}$.*

Apart from absolute constants, the lower and upper bounds differ by a factor of $1/\sqrt{d}$.[7] We point out that the $1/\sqrt{d}$ factor is not improvable unless using functions more sophisticated than Lipschitz functions of one-dimensional projections of $x$. Indeed, $W_{\mathcal{F}}(p,q)$ is upper bounded by the maximum Wasserstein distance between one-dimensional projections of $p, q$, which is on the order of $W_1(p,q)/\sqrt{d}$ when $p, q$ have spherical covariances. The proof is deferred to Section B.1.

**Extension to mixture of Gaussians.** Discriminator family $\mathcal{F}$ with restricted approximability can also be designed for mixture of Gaussians. We defer this result and the proof to Appendix C.

---

[5]This was defined as neural net distance in (Arora et al., 2017a).

[6]In the ideal case we can take the expectation over $q$, as the generator $q$ is able to generate infinitely many samples.

[7]As shown in (Feizi et al., 2017), the optimal discriminator for Gaussian distributions are quadratic functions.

## 3.2 EXPONENTIAL FAMILIES

Now we consider exponential families and show that the linear combinations of the sufficient statistics are a family of discriminators with restricted approximability. Concretely, let $\mathcal{G} = \{p_\theta : \theta \in \Theta \subset \mathbb{R}^k\}$ be an exponential family, where $p_\theta(x) = \frac{1}{Z(\theta)} \exp(\langle \theta, T(x) \rangle)$, $\forall x \in \mathcal{X} \subset \mathbb{R}^d$: here $T : \mathbb{R}^d \to \mathbb{R}^k$ is the vector of sufficient statistics, and $Z(\theta)$ is the partition function. Let the discriminator family be all linear functionals over the features $T(x)$: $\mathcal{F} = \{x \to \langle v, T(x) \rangle : \|v\|_2 \leq 1\}$.

**Theorem 3.2.** *Let $\mathcal{G}$ be the exponential family and $\mathcal{F}$ be the discriminators defined above. Assume that the log partition function $\log Z(\theta)$ satisfies that $\gamma I \preceq \nabla^2 \log Z(\theta) \preceq \beta I$. Then we have for any $p, q \in \mathcal{G}$,*

$$\frac{\gamma}{\sqrt{\beta}} \sqrt{D_{\mathrm{kl}}(p\|q)} \leq W_{\mathcal{F}}(p,q) \leq \frac{\beta}{\sqrt{\gamma}} \sqrt{D_{\mathrm{kl}}(p\|q)}. \tag{7}$$

*If we further assume $\mathcal{X}$ has diameter $D$ and $T(x)$ is L-Lipschitz in $\mathcal{X}$. Then,*

$$\frac{D\gamma}{\sqrt{\beta}} W_1(p,q) \lesssim W_{\mathcal{F}}(p,q) \leq L \cdot W_1(p,q) \tag{8}$$

*Moreover, $\mathcal{F}$ has Rademacher complexity bound $R_n(\mathcal{F}, \mathcal{G}) \leq \sqrt{\frac{\sup_{\theta \in \Theta} E_{p_\theta}[\|T(X)\|_2^2]}{n}}$.*

We note that the log partition function $\log Z(\theta)$ is always convex, and therefore our assumptions only require in addition that the curvature (i.e. the Fisher information matrix) has a strictly positive lower bound and a global upper bound. For the bound eq. (8), some geometric assumptions on the sufficient statistics are necessary because the Wasserstein distance intrinsically depends on the underlying geometry of $x$, which are not specified in exponential families by default. The proof of eq. (7) follows straightforwardly from the standard theory of exponential families. The proof of eq. (8) requires machinery that we will develop in Section 4 and is therefore deferred to Section B.2.

# 4 RESTRICTED APPROXIMABILITY FOR NEURAL NET GENERATORS

In this section, we design discriminators with restricted approximability for neural net generators, a family of distributions that are widely used in GANs to model real data.

In Section 4.1 we consider the invertible neural networks generators which have proper densities. In Section 4.2, we extend the results to the more general and challenging setting of injective neural networks generators, where the latent variables are allowed to have lower dimension than the observable dimensions (Theorem 4.5) and the distributions no longer have densities.

## 4.1 INVERTIBLE NEURAL NETWORK GENERATORS

In this section, we consider the generators that are parameterized by invertible neural networks[8]. Concretely, let $\mathfrak{G}$ be a family of neural networks $\mathfrak{G} = \{G_\theta : \theta \in \Theta\}$. Let $p_\theta$ be the distribution of

$$X = G_\theta(Z), \ Z \sim \mathsf{N}(0, \mathrm{diag}(\gamma^2)). \tag{9}$$

where $G_\theta$ is a neural network with parameters $\theta$ and $\gamma \in \mathbb{R}^d$ standard deviation of hidden factors. By allowing the variances to be non-spherical, we allow each hidden dimension to have a different impact on the output distribution. In particular, the case $\gamma = [\mathbf{1}_k, \delta\mathbf{1}_{d-k}]$ for some $\delta \ll 1$ has the ability to model data around a "$k$-dimensional manifold" with some noise on the level of $\delta$.

We are interested in the set of invertible neural networks $G_\theta$. We let our family $\mathcal{G}$ consist of standard $\ell$-layer feedforward nets $x = G_\theta(z)$ of the form

$$x = W_\ell \sigma(W_{\ell-1} \sigma(\cdots \sigma(W_1 z + b_1) \cdots) + b_{\ell-1}) + b_\ell,$$

where $W_i \in \mathbb{R}^{d \times d}$ are invertible, $b_i \in \mathbb{R}^d$, and $\sigma : \mathbb{R} \to \mathbb{R}$ is the activation function, on which we make the following assumption:

---

[8]Our techniques also applies to other parameterized invertible generators but for simplicity we only focus on neural networks.

**Assumption 1** (Invertible generators). *Let $R_W, R_b, \kappa_\sigma, \beta_\sigma > 0, \delta \in (0, 1]$ be parameters which are considered as constants (that may depend on the dimension). We consider neural networks $G_\theta$ that are parameterized by parameters $\theta = (W_i, b_i)_{i \in [\ell]}$ belonging to the set*

$$\Theta = \left\{ (W_i, b_i)_{i \in [\ell]} : \max\left\{ \|W_i\|_{\mathrm{op}}, \|W_i^{-1}\|_{\mathrm{op}} \right\} \leq R_W, \ \|b_i\|_2 \leq R_b, \ \forall i \in [\ell] \right\}.$$

*The activation function $\sigma$ is twice-differentiable with $\sigma(0) = 0$, $\sigma'(t) \in [\kappa_\sigma^{-1}, 1]$, and $|(\sigma^{-1})''/(\sigma^{-1})'| \leq \beta_\sigma$. The standard deviation of the hidden factors satisfy $\gamma_i \in [\delta, 1]$.*

Clearly, such a neural net is invertible, and its inverse is also a feedforward neural net with activation $\sigma^{-1}$. We note that a smoothed version of Leaky ReLU (Xu et al., 2015) satisfies all the conditions on the activation functions. Further, it is necessary to impose some assumptions on the generator networks because arbitrary neural networks are likely to be able to implement pseudo-random functions which can't be distinguished from random functions by even any polynomial time algorithms.

**Lemma 4.1.** *For any $\theta \in \Theta$, the function $\log p_\theta$ can be computed by a neural network with at most $\ell + 1$ layers, $O(\ell d^2)$ parameters, and activation function among $\{\sigma^{-1}, \log \sigma^{-1'}, (\cdot)^2\}$ of the form*

$$f_\phi(x) = \frac{1}{2} \left\langle h_1, \mathrm{diag}(\gamma^{-2}) h_1 \right\rangle + \sum_{k=2}^{\ell} \left\langle \mathbf{1}_d, \log \sigma^{-1'}(h_j) \right\rangle + C, \tag{10}$$

*where $h_\ell = W_\ell(x - b_\ell)$, $h_k = W_k(\sigma^{-1}(h_{k+1}) - b_k)$ for $k \in \{\ell - 1, \ldots, 1\}$, and the parameter $\phi = ((W_j, b_j)_{j=1}^{\ell}, C)$ satisfies $\phi \in \Phi = \{\phi : \|W_j\|_{\mathrm{op}} \leq R_W, \ \|b_j\|_2 \leq R_b, \ |C| \leq (\ell - 1)d \log R_W\}$. As a direct consequence, the following family $\mathcal{F}$ of neural networks with activation functions above of at most $\ell + 2$ layers contains all the functions $\{\log p - \log q : p, q \in \mathcal{G}\}$ :*

$$\mathcal{F} = \{f_{\phi_1} - f_{\phi_2} : \phi_1, \phi_2 \in \Phi\}. \tag{11}$$

We note that the exact form of the parameterized family $\mathcal{F}$ is likely not very important in practice, since other family of neural nets also possibly contain good approximations of $\log p - \log q$ (which can be seen partly from experiments in Section G.)

The proof builds on the change-of-variable formula $\log p_\theta(x) = \log \phi_\gamma(G_\theta^{-1}(x)) + \log |\det \frac{\partial G_\theta^{-1}(x)}{\partial x}|$ (where $\phi_\gamma$ is the density of $Z \sim \mathsf{N}(0, \mathrm{diag}(\gamma^2))$) and the observation that $G_\theta^{-1}$ is a feedforward neural net with $\ell$ layers. Note that the log-det of the Jacobian involves computing the determinant of the (inverse) weight matrices. A priori such computation is non-trivial for a given $G_\theta$. However, it's just some constant that does not depend on the input, therefore it can be representable by adding a bias on the final output layer. This frees us from further structural assumptions on the weight matrices (in contrast to the architectures in flow-GANs (Gulrajani et al., 2017)). We defer the proof of Lemma 4.1 to Section D.2.

**Theorem 4.2.** *Suppose $\mathcal{G} = \{p_\theta : \theta \in \Theta\}$ is the set of invertible-generator distributions as defined in eq. (9) satisfying Assumption 1. Then, the discriminator class $\mathcal{F}$ defined in Lemma 4.1 has restricted approximability w.r.t. $\mathcal{G}$ in the sense that for any $p, q \in \mathcal{G}$,*

$$W_1(p, q)^2 \lesssim D_{\mathrm{kl}}(p\|q) + D_{\mathrm{kl}}(q\|p) \leq W_{\mathcal{F}}(p, q) \lesssim \frac{\sqrt{d}}{\delta^2} \left( W_1(p, q) + d \exp(-10d) \right),$$

*When $n \gtrsim \max\left\{ d, \delta^{-8} \log 1/\delta \right\}$, we have the generalization bound $R_n(\mathcal{F}, \mathcal{G}) \leq \varepsilon_{\mathrm{gen}} := \sqrt{\frac{d^4 \log n}{\delta^4 n}}$.*

The proof of Theorem 4.2 uses the following lemma that relates the KL divergence to the IPM when the log densities exist and belong to the family of discriminators.

**Lemma 4.3** (Special case of (Zhang et al., 2017, Proposition 2.9)). *Let $\varepsilon > 0$. Suppose $\mathcal{F}$ satisfies that for every $q \in \mathcal{G}$, there exists $f \in \mathcal{F}$ such that $\|f - (\log p - \log q)\|_\infty \leq \epsilon$, and that all the functions in $\mathcal{F}$ are $L$-Lipschitz. Then,*

$$D_{\mathrm{kl}}(p\|q) + D_{\mathrm{kl}}(q\|p) - \varepsilon \leq W_{\mathcal{F}}(p, q) \leq L \cdot W_1(p, q). \tag{12}$$

We outline a proof sketch of Theorem 4.2 below and defer the full proof to Appendix D.3. As we choose the discriminator class as in Lemma 4.1 which implements $\log p - \log q$ for any $p, q \in \mathcal{G}$,

by Lemma 4.3, $W_{\mathcal{F}}(p, q)$ is lower bounded by $D_{\mathrm{kl}}(p\|q) + D_{\mathrm{kl}}(q\|p)$. It thus suffices to (1) lower bound this quantity by the Wasserstein distance and (2) upper bound $W_{\mathcal{F}}(p, q)$ by the Wasserstein distance.

To establish (1), we will prove in Lemma D.3 that for any $p, q \in \mathcal{G}$,

$$W_1(p, q)^2 \leq W_2(p, q)^2 \lesssim D_{\mathrm{kl}}(p\|q) + D_{\mathrm{kl}}(q\|p).$$

Such a result is the simple implication of *transportation inequalities* by Bobkov-Götze and Gozlan (Theorem D.1), which state that if $X \sim p$ (or $q$) and $f$ is 1-Lipschitz implies that $f(X)$ is sub-Gaussian, then the inequality above holds. In our invertible generator case, we have $X = G_\theta(Z)$ where $Z$ are independent Gaussians, so as long as $G_\theta$ is suitably Lipschitz, $f(X) = f(G_\theta(Z))$ is a sub-Gaussian random variable by the standard Gaussian concentration result (Vershynin, 2010).

The upper bound (2) would have been immediate if functions in $\mathcal{F}$ are Lipschitz globally in the whole space. While this is not strictly true, we give two workarounds – by either doing a truncation argument to get a $W_1$ bound with some tail probability, or a $W_2$ bound which only requires the Lipschitz constant to grow at most linearly in $\|x\|_2$. This is done in Theorem D.2 as a straightforward extension of the result in (Polyanskiy & Wu, 2016).

Combining the restricted approximability and the generalization bound, we immediately obtain that if the training succeeds with small expected IPM (over the randomness of the learned distributions), then the estimated distribution $q$ is close to the true distribution $p$ in Wasserstein distance.

**Corollary 4.4.** *In the setting of Theorem 4.2, with high probability over the choice of training data $\hat{p}^n$, we have that if the training process returns a distribution $q \in \mathcal{G}$ such that $\mathbb{E}_{\hat{q}^n}[W_{\mathcal{F}}(\hat{p}^n, \hat{q}^n)] \leq \varepsilon_{\mathrm{train}}$, then with $\varepsilon_{\mathrm{gen}} := \sqrt{\frac{d^4 \log n}{\delta^4 n}}$, we have*

$$W_1(p, q) \leq W_2(p, q) \lesssim (\varepsilon_{\mathrm{train}} + \varepsilon_{\mathrm{gen}})^{1/2}. \tag{13}$$

We note that the training error is measured by $\mathbb{E}_{\hat{q}^m}[W_{\mathcal{F}}(\hat{p}^n, \hat{q}^m)]$, the expected IPM over the randomness of the learned distributions, which is a measurable value because one can draw fresh samples from $q$ to estimate the expectation. It's an important open question to design efficient algorithms to achieve a small training error according to this definition, and this is left for future work.

## 4.2 Injective neural network generators

In this section we consider injective neural network generators (defined below) which generate distributions residing on a low dimensional manifold. This is a more realistic setting than Section 4.1 for modeling real images, but technically more challenging because the KL divergence becomes infinity, rendering Lemma 4.3 useless. Nevertheless, we design a novel divergence between two distributions that is sandwiched by Wasserstein distance and can be optimized as IPM.

Concretely, we consider a family of neural net generators $\mathfrak{G} = \{G_\theta : \mathbb{R}^k \to \mathbb{R}^d\}$ where $k < d$ and $G_\theta$ is injective function. [9] Therefore, $G_\theta$ is invertible only on the image of $G_\theta$, which is a $k$-dimensional manifold in $\mathbb{R}^d$. Let $\mathcal{G}$ be the corresponding family of distributions generated by neural nets in $\mathfrak{G}$.

Our key idea is to design a variant of the IPM, which provably approximates the Wasserstein distance. Let $p^\beta$ denote the convolution of the distribution $p$ with a Gaussian distribution $\mathsf{N}(0, \beta^2 I)$. We define a smoothed $\mathcal{F}$-IPM between $p, q$ as

$$\tilde{d}_{\mathcal{F}}(p, q) \triangleq \inf_{\beta \geq 0}\ (W_{\mathcal{F}}(p^\beta, q^\beta) + \beta \log 1/\beta)^{1/2}, \tag{14}$$

Clearly $\tilde{d}_{\mathcal{F}}$ can be optimized as $W_{\mathcal{F}}$ with an additional variable $\beta$ introduced in the optimization. We show that for certain discriminator class (see Section E for the details of the construction) such that $\tilde{d}_{\mathcal{F}}$ approximates the Wasserstein distance.

**Theorem 4.5** (Informal version of Theorem E.1). *Let $\mathcal{G}$ be defined as above. There exists a discriminator class $\mathcal{F}$ such that for any pair of distributions $p, q \in \mathcal{G}$, we have*

$$W_1(p, q) \lesssim \tilde{d}_{\mathcal{F}}(p, q) \lesssim \mathrm{poly}(d) \cdot W_1(p, q)^{1/6} + \exp(-\Omega(d)). \tag{15}$$

---

[9] In other words, $G_\theta(x) \neq G_\theta(y)$ if $x \neq y$.

*Furthermore, when $n \gtrsim \text{poly}(d)$, we have the generalization bound*

$$R_n(\mathcal{F}, \mathcal{G}) \lesssim \text{poly}(d)\sqrt{\frac{\log n}{n}}$$

*Here* $\text{poly}(d)$ *hides polynomial dependencies on* $d$ *and several other parameters that will be defined in the formal version (Theorem E.1.)*

The direct implication of the theorem is that if $\tilde{d}(\hat{p}^n, \hat{q}^n)$ is small for $n \gtrsim \text{poly}(d)$, then $W(p, q)$ is guaranteed to be also small and thus we don't have mode collapse.

## 5 SIMULATION

Our theoretical results on neural network generators in Section 4 convey the message that mode collapse will not happen as long as the discriminator family $\mathcal{F}$ has restricted approximability with respect to the generator family $\mathcal{G}$. In particular, the IPM $W_{\mathcal{F}}(p, q)$ is upper and lower bounded by the Wasserstein distance $W_1(p, q)$ given the restricted approximability. We design certain specific discriminator classes in our theory to guarantee this, but we suspect it holds more generally in GAN training in practice.

We perform two sets of synthetic experiments to confirm that the practice is indeed consistent with our theory. We design synthetic datasets, set up suitable generators, and train GANs with either our theoretically proposed discriminator class with restricted approximability, or vanilla neural network discriminators of reasonable capacity. In both cases, we show that IPM is well correlated with the Wasserstein / KL divergence, suggesting that the restricted approximability may indeed hold in practice. This suggests that the difficulty of GAN training in practice may come from the optimization difficulty rather than statistical inefficiency, as we observe evidence of good statistical behaviors on "typcial" discriminator classes.

We briefly describe the experiments here and defer details of the second experiment to Appendix G.

(a) We learn synthetic 2D datasets with neural net generators and discriminators and show that the IPM is well-correlated with the Wasserstein distance (Section 5.1).

(b) We learn invertible neural net generators with discriminators of restricted approximability and vanilla architectures (Appendix G). We show that the IPM is well-correlated with the KL divergence, both along training and when we consider two generators that are perturbations of each other (the purpose of the latter being to eliminate any effects of the optimization).

### 5.1 EXPERIMENTS ON SYNTHETIC 2D DATASETS

In this section, we perform synthetic experiments with WGANs that learn various curves in two dimensions. In particular, we will train GANs that learn the unit circle and a "swiss roll" curve (Gulrajani et al., 2017) – both distributions are supported on a one-dimensional manifold in $\mathbb{R}^2$, therefore the KL divergence does not exist, but one can use the Wasserstein distance to measure the quality of the learned generator.

We show that WGANs are able to learn both distributions pretty well, and the IPM $W_{\mathcal{F}}$ is strongly correlated with the Wasserstein distance $W_1$. These ground truth distributions are not covered in our Theorems 4.2 and 4.5, but our results show evidence that restricted approximability is still quite likely to hold here.

**Ground truth distributions** We set the ground truth distribution to be a unit circle or a Swiss roll curve, sampled from

$$\text{Circle}: \ (x, y) \sim \text{Uniform}(\{(x, y) : x^2 + y^2 = 1\})$$
$$\text{Swiss roll}: \ (x, y) = (z \cos(4\pi z), z \sin(4\pi z)) : \ z \sim \text{Uniform}([0.25, 1]).$$

**Generators and discriminators** We use standard two-hidden-layer ReLU nets as both the generator class and the discriminator class. The generator architecture is 2-50-50-2, and the discriminator architecture is 2-50-50-1. We use the RMSProp optimizer (Tieleman & Hinton, 2012) as our update

rule, the learning rates are $10^{-4}$ for both the generator and discriminator, and we perform 10 steps on the discriminator in between each generator step.

**Metric** We compare two metrics between the ground truth distribution $p$ and the learned distribution $q$ along training:

(1) The neural net IPM $W_{\mathcal{F}}(p, q)$, computed on fresh batches from $p, q$ through optimizing a separate discriminator from cold start.

(2) The Wasserstein distance $W_1(p, q)$, computed on fresh batches from $p, q$ using the POT package[10]. As data are in two dimensions, the empirical Wasserstein distance $W_1(\hat{p}, \hat{q})$ does not suffer from the curse of dimensionality and is a good proxy of the true Wasserstein distance $W_1(p, q)$ (Weed & Bach, 2017).

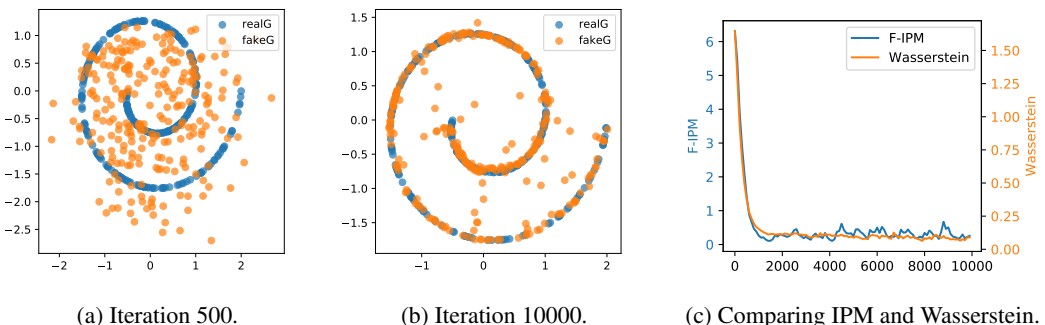

(a) Iteration 500.  (b) Iteration 10000.  (c) Comparing IPM and Wasserstein.

Figure 1: Experiments on the swiss roll dataset. The neural net IPM, the Wasserstein distance, and the sample quality are correlated along training. (a)(b): Sample batches from the ground truth and the learned generator at iteration 500 and 5000. (c): Comparing the F-IPM and the Wasserstein distance. RealG and fakeG denote the ground truth generator and the learned generator, respectively.

**Result** See Figure 1 for the Swiss roll experiment and Figure 2 (in Appendix F) for the unit circle experiment. On both datasets, the learned generator is very close to the ground truth distribution at iteration 10000. Furthermore, the neural net IPM and the Wasserstein distance are well correlated. At iteration 500, the generators have not quite learned the true distributions yet (by looking at the sampled batches), and the IPM and Wasserstein distance are indeed large.

## 6 CONCLUSION

We present the first polynomial-in-dimension sample complexity bounds for learning various distributions (such as Gaussians, exponential families, invertible neural networks generators) using GANs with convergence guarantees in Wasserstein distance (for distributions with low-dimensional supports) or KL divergence. The analysis technique proceeds via designing discriminators with restricted approximability – a class of discriminators tailored to the generator class in consideration which have good generalization and mode collapse avoidance properties.

We hope our techniques can be in future extended to other families of distributions with tighter sample complexity bounds. This would entail designing discriminators that have better restricted approximability bounds, and generally exploring and generalizing approximation theory results in the context of GANs. We hope such explorations will prove as rich and satisfying as they have been in the vanilla functional approximation settings.

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

## A    PROOFS FOR SECTION 2

### A.1    PROOF OF THEOREM 2.1

Fixing $\hat{p}^n$, consider a random sample $\hat{q}^n$. It is easy to verify that the $\mathcal{F}$-IPM satisfies the triangle inequality, so we have

$$|W_{\mathcal{F}}(p,q) - \mathbb{E}_{\hat{q}^n}[W_{\mathcal{F}}(\hat{p}^n, \hat{q}^n)]| \leq \mathbb{E}_{\hat{q}^n}[|W_{\mathcal{F}}(p,q) - W_{\mathcal{F}}(\hat{p}^n, \hat{q}^n)]$$
$$\leq \mathbb{E}_{\hat{q}^n}[W_{\mathcal{F}}(p,\hat{p}^n) + W_{\mathcal{F}}(q,\hat{q}^n)] = W_{\mathcal{F}}(p,\hat{p}^n) + \mathbb{E}_{\hat{q}^n}[W_{\mathcal{F}}(q,\hat{q}^n)].$$

Taking expectation over $\hat{p}^n$ on the above bound yields

$$\mathbb{E}_{\hat{p}^n}\left[|W_{\mathcal{F}}(p,q) - \mathbb{E}_{\hat{q}^n}[W_{\mathcal{F}}(\hat{p}^n, \hat{q}^n)]|\right] \leq \mathbb{E}_{\hat{p}^n}[W_{\mathcal{F}}(p,\hat{p}^n)] + \mathbb{E}_{\hat{q}^n}[W_{\mathcal{F}}(q,\hat{q}^n)].$$

So it suffices to bound $\mathbb{E}_{\hat{p}^n}[W_{\mathcal{F}}(p,\hat{p}^n)]$ by $2R_n(\mathcal{F},\mathcal{G})$ and the same bound will hold for $q$. Let $X_i$ be the samples in $\hat{p}^n$. By symmetrization, we have

$$W_{\mathcal{F}}(p,\hat{p}^n) = \mathbb{E}\left[\sup_{f\in\mathcal{F}}\left|\frac{1}{n}\sum_{i=1}^{n}f(X_i) - \mathbb{E}_p[f(X)]\right|\right] \leq 2\mathbb{E}\left[\sup_{f\in\mathcal{F}}\left|\frac{1}{n}\sum_{i=1}^{n}\varepsilon_i f(X_i)\right|\right] = 2\mathbb{E}[R_n(\mathcal{F},p)] \leq 2R_n(\mathcal{F},\mathcal{G}).$$

Adding up this bound and the same bound for $q$ gives the desired result.

## B    PROOFS FOR SECTION 3

### B.1    PROOF OF THEOREM 3.1

Recall that our discriminator family is

$$\mathcal{F} = \left\{x \mapsto \sigma(v^\top x + b) : \|v\|_2 \leq 1, |b| \leq D\right\}.$$

**Restricted approximability**    The upper bound $W_{\mathcal{F}}(p_1, p_2) \leq W_1(p_1, p_2)$ follows directly from the fact that functions in $\mathcal{F}$ are 1-Lipschitz.

We now establish the lower bound. First, we recover the mean distance, in which we use the following simple fact: a linear discriminator is the sum of two ReLU discriminators, or mathematically $t = \sigma(t) - \sigma(-t)$. Taking $v = \frac{\mu_1 - \mu_2}{\|\mu_1 - \mu_2\|_2}$, we have

$$\|\mu_1 - \mu_2\|_2 = v^\top \mu_1 - v^\top \mu_2 = \mathbb{E}_{p_1}[v^\top X] - \mathbb{E}_{p_2}[v^\top X]$$
$$= \left(\mathbb{E}_{p_1}[\sigma(v^\top X)] - \mathbb{E}_{p_2}[\sigma(v^\top X)]\right) + \left(-\mathbb{E}_{p_1}[\sigma(-v^\top X)] + \mathbb{E}_{p_2}[\sigma(-v^\top X)]\right)$$
$$\leq \left|\mathbb{E}_{p_1}[\sigma(v^\top X)] - \mathbb{E}_{p_2}[\sigma(v^\top X)]\right| + \left|\mathbb{E}_{p_1}[\sigma(-v^\top X)] - \mathbb{E}_{p_2}[\sigma(-v^\top X)]\right|.$$

Therefore at least one of the above two terms is greater than $\|\mu_1 - \mu_2\|_2/2$, which shows that $W_{\mathcal{F}}(p_1, p_2) \geq \|\mu_1 - \mu_2\|_2/2$.

For the covariance distance, we need to actually compute $\mathbb{E}_p[\sigma(v^\top X + b)]$ for $p = \mathsf{N}(\mu, \Sigma)$. Note that $X \stackrel{d}{=} \Sigma^{1/2}Z + \mu$, where $Z \sim \mathsf{N}(0, I_d)$. Further, we have $v^\top X \stackrel{d}{=} \|\Sigma^{1/2}v\|_2 W + v^\top \mu$ for $W \sim \mathsf{N}(0,1)$, therefore

$$\mathbb{E}_p[\sigma(v^\top X + b)] = \mathbb{E}\left[\sigma\left(\|\Sigma^{1/2}v\|_2 W + v^\top \mu + b\right)\right]$$
$$= \|\Sigma^{1/2}v\|_2 \mathbb{E}\left[\sigma\left(W + \frac{v^\top \mu + b}{\|\Sigma^{1/2}v\|_2}\right)\right] = \|\Sigma^{1/2}v\|_2 R\left(\frac{v^\top \mu + b}{\|\Sigma^{1/2}v\|_2}\right).$$

(Defining $R(a) = \mathbb{E}[\max\{W + a, 0\}]$ for $W \sim \mathsf{N}(0,1)$.) Therefore, the neuron distance between the two Gaussians is

$$W_{\mathcal{F}}(p_1, p_2) = \sup_{\|v\|_2 \leq 1, |b| \leq D}\left|\|\Sigma_1^{1/2}v\|_2 R\left(\frac{v^\top \mu_1 + b}{\|\Sigma_1^{1/2}v\|_2}\right) - \|\Sigma_2^{1/2}v\|_2 R\left(\frac{v^\top \mu_2 + b}{\|\Sigma_2^{1/2}v\|_2}\right)\right|,$$

As $a \mapsto \max\{a + w, 0\}$ is strictly increasing for all $w$, the function $R$ is strictly increasing. It is also a basic fact that $R(0) = 1/\sqrt{2\pi}$.

Consider any fixed $v$. By flipping the sign of $v$, we can let $v^\top \mu_1 \geq v^\top \mu_2$ without changing $\left\| \Sigma_i^{1/2} v \right\|_2$. Now, letting $b = -v^\top(\mu_1 - \mu_2)/2$ (note that $|b| \leq D$ is a valid choice), we have

$$v^\top \mu_1 + b = \frac{v^\top(\mu_1 - \mu_2)}{2} \geq 0, \quad v^\top \mu_2 + b = -\frac{v^\top(\mu_1 - \mu_2)}{2} \leq 0.$$

As $R$ is strictly increasing, for this choice of $(v, b)$ we have

$$\left\| \Sigma_1^{1/2} v \right\|_2 R\left( \frac{v^\top \mu_1 + b}{\left\| \Sigma_1^{1/2} v \right\|_2} \right) - \left\| \Sigma_2^{1/2} v \right\|_2 R\left( \frac{v^\top \mu_2 + b}{\left\| \Sigma_2^{1/2} v \right\|_2} \right)$$

$$\geq R(0) \left( \left\| \Sigma_1^{1/2} v \right\|_2 - \left\| \Sigma_2^{1/2} v \right\|_2 \right) = \frac{1}{\sqrt{2\pi}} \left( \left\| \Sigma_1^{1/2} v \right\|_2 - \left\| \Sigma_2^{1/2} v \right\|_2 \right).$$

Ranging over $\|v\|_2 \leq 1$ we then have

$$W_\mathcal{F}(p_1, p_2) \geq \frac{1}{\sqrt{2\pi}} \sup_{\|v\|_2 \leq 1} \left| \left\| \Sigma_1^{1/2} v \right\|_2 - \left\| \Sigma_2^{1/2} v \right\|_2 \right|.$$

The quantity in the supremum can be further bounded as

$$\left| \left\| \Sigma_1^{1/2} v \right\|_2 - \left\| \Sigma_2^{1/2} v \right\|_2 \right| = \frac{|v^\top(\Sigma_1 - \Sigma_2)v|}{\left\| \Sigma_1^{1/2} v \right\|_2 + \left\| \Sigma_2^{1/2} v \right\|_2} \geq \frac{|v^\top(\Sigma_1 - \Sigma_2)v|}{\lambda_{\max}(\Sigma_1^{1/2}) + \lambda_{\max}(\Sigma_2^{1/2})}.$$

Choosing $v = v_{\max}(\Sigma_1 - \Sigma_2)$ gives

$$W_\mathcal{F}(p_1, p_2) \geq \frac{1}{\sqrt{2\pi}} \sup_{\|v\|_2 \leq 1} \left| \left\| \Sigma_1^{1/2} v \right\|_2 - \left\| \Sigma_2^{1/2} v \right\|_2 \right| \geq \frac{\|\Sigma_1 - \Sigma_2\|_{\mathrm{op}}}{\sqrt{2\pi} 2\sigma_{\max}}.$$

Now, using the perturbation bound

$$\left\| \Sigma_1^{1/2} - \Sigma_2^{1/2} \right\|_{\mathrm{op}} \leq \frac{1}{\lambda_{\min}(\Sigma_1) + \lambda_{\min}(\Sigma_2)} \cdot \|\Sigma_1 - \Sigma_2\|_{\mathrm{op}} \leq \frac{1}{2\sigma_{\min}} \|\Sigma_1 - \Sigma_2\|_{\mathrm{op}},$$

(cf. (Schmitt, 1992, Lemma 2.2)), we get

$$W_\mathcal{F}(p_1, p_2) \geq \frac{1}{2\sqrt{2\pi}\sigma_{\max}} \cdot 2\sigma_{\min} \left\| \Sigma_1^{1/2} - \Sigma_2^{1/2} \right\|_{\mathrm{op}} \geq \frac{\sigma_{\min}}{\sqrt{2\pi}\sigma_{\max}\sqrt{d}} \left\| \Sigma_1^{1/2} - \Sigma_2^{1/2} \right\|_{\mathrm{Fr}}.$$

Combining the above bound with the bound in the mean difference, we get

$$W_\mathcal{F}(p_1, p_2) \geq \frac{1}{2} \left( \frac{\|\mu_1 - \mu_2\|_2}{2} + \frac{\sigma_{\min}}{\sqrt{2\pi d}\sigma_{\max}} \left\| \Sigma_1^{1/2} - \Sigma_2^{1/2} \right\|_{\mathrm{Fr}} \right)$$

$$\geq \frac{\sigma_{\min}}{2\sqrt{2\pi d}\sigma_{\max}} \sqrt{\|\mu_1 - \mu_2\|_2^2 + \inf_{U^\top U = UU^\top = I_d} \left\| \Sigma_1^{1/2} - U\Sigma_2^{1/2} \right\|_{\mathrm{Fr}}^2}$$

$$= \frac{\sigma_{\min}}{2\sqrt{2\pi d}\sigma_{\max}} \cdot W_2(p_1, p_2) \geq \frac{\sigma_{\min}}{2\sqrt{2\pi d}\sigma_{\max}} \cdot W_1(p_1, p_2) \qquad (16)$$

The last equality following directly from the closed-form expression of the $W_2$ distance between two Gaussians (Masarotto et al., 2018, Proposition 3). Thus the claimed lower bound holds with $c = 1/(2\sqrt{2\pi})$.

**KL Bound** We use the $W_2$ distance to bridge the KL and the $\mathcal{F}$-distance, which uses the machinery developed in Section D. Let $p_1, p_2$ be two Gaussians distributions with parameters $\theta_i = (\mu_i, \Sigma_i) \in \Theta$. By the equality

$$D_{\mathrm{kl}}(p_1 \| p_2) + D_{\mathrm{kl}}(p_2 \| p_1) = (\mathbb{E}_{p_1}[\log p_1(X)] - \mathbb{E}_{p_2}[\log p_1(X)]) + (\mathbb{E}_{p_2}[\log p_2(X)] - \mathbb{E}_{p_1}[\log p_2(X)]),$$

it suffices to upper bound the term only involving $\log p_1(X)$ (the other follows similarly), which by Theorem D.2 requires bounding the growth of $\|\nabla \log p_1(x)\|_2$. We have

$$\|\nabla \log p_1(x)\|_2 = \left\|\Sigma_1^{-1}(x - \mu_1)\right\|_2 \le \sigma_{\min}^{-2}\|x - \mu_1\|_2.$$

Further $\mathbb{E}_{p_i}[\|x - \mu_1\|_2^2] \le \operatorname{tr}(\Sigma_i) + \|\mu_i - \mu_1\|_2^2 \le d\sigma_{\max}^2 + 4D^2$ for $i = 1, 2$, therefore by (a trivial variant of) Theorem D.2(c) we get

$$\mathbb{E}_{p_1}[\log p_1(X)] - \mathbb{E}_{p_2}[\log p_1(X)] \le \sigma_{\min}^{-2}(\sqrt{d}\sigma_{\max} + 2D)W_2(p_1, p_2).$$

The same bound holds for $\log p_2$. Adding them up and substituting the bound appendix B.1 gives that

$$D_{\mathrm{kl}}(p_1\|p_2) + D_{\mathrm{kl}}(p_2\|p_1) \lesssim \frac{\sqrt{d}\sigma_{\max} + 2D}{\sigma_{\min}^2}W_2(p_1, p_2) \lesssim \frac{\sqrt{d}\sigma_{\max}(\sqrt{d}\sigma_{\max} + D)}{\sigma_{\min}^3}W_{\mathcal{F}}(p_1, p_2).$$

**Generalization** We wish to bound for all $\theta = (\mu, \Sigma) \in \Theta$

$$R_n(\mathcal{F}, p_\theta) = \mathbb{E}_{p_\theta}\left[\sup_{\|v\|_2 \le 1, |b| \le D}\left|\frac{1}{n}\sum_{i=1}^n \varepsilon_i\sigma(v^\top X_i + b)\right|\right].$$

As $\sigma : \mathbb{R} \to \mathbb{R}$ is 1-Lipschitz, by the Rademacher contraction inequality (Ledoux & Talagrand, 2013), we have

$$\mathbb{E}_{p_\theta}\left[\sup_{\|v\|_2 \le 1, |b| \le D}\left|\frac{1}{n}\sum_{i=1}^n \varepsilon_i\sigma(v^\top X_i + b)\right|\right] \le 2\mathbb{E}_{p_\theta}\left[\sup_{\|v\|_2 \le 1, |b| \le D}\left|\frac{1}{n}\sum_{i=1}^n \varepsilon_i(v^\top X_i + b)\right|\right].$$

The right hand side can be bounded directly as

$$\mathbb{E}_{p_\theta}\left[\sup_{\|v\|_2 \le 1, |b| \le D}\left|\frac{1}{n}\sum_{i=1}^n \varepsilon_i(v^\top X_i + b)\right|\right] = E_{p_\theta}\left[\sup_{\|v\|_2 \le 1, |b| \le D}\left|(b + v^\top\mu)\frac{1}{n}\sum_{i=1}^n \varepsilon_i + \frac{1}{n}\sum_{i=1}^n \varepsilon_i v^\top(X_i - \mu)\right|\right]$$

$$\le \sup_{\|v\|_2 \le 1, |b| \le D}|b + v^\top\mu|\mathbb{E}\left[\left|\frac{1}{n}\sum_{i=1}^n \varepsilon_i\right|\right] + \mathbb{E}_{p_\theta}\left[\sup_{\|v\|_2 \le 1}\left|\left\langle v, \frac{1}{n}\sum_{i=1}^n \varepsilon_i(X_i - \mu)\right\rangle\right|\right]$$

$$\le 2D\mathbb{E}\left[\left|\frac{1}{n}\sum_{i=1}^n \varepsilon_i\right|\right] + \mathbb{E}_{p_\theta}\left[\left\|\frac{1}{n}\sum_{i=1}^n \varepsilon_i(X_i - \mu)\right\|_2\right]$$

$$\le 2D\sqrt{\mathbb{E}\left[\left(\frac{1}{n}\sum_{i=1}^n \varepsilon_i\right)^2\right]} + \sqrt{\mathbb{E}_{p_\theta}\left[\left\|\frac{1}{n}\sum_{i=1}^n \varepsilon_i(X_i - \mu)\right\|_2^2\right]}$$

$$= \frac{2D + \sqrt{\operatorname{tr}(\Sigma)}}{\sqrt{n}} \le \frac{2D + \sigma_{\max}\sqrt{d}}{\sqrt{n}}.$$

## B.2 Proof of Theorem 3.2

**KL bounds** Recall the basic property of exponential family that $A(\theta) = \mathbb{E}_{p_\theta}[T(X)]$. Suppose $p = p_{\theta_1}$ and $q = p_{\theta_2}$. Then,

$$W_{\mathcal{F}}(p, q) = \sup_{\|v\|_2 \le 1}\mathbb{E}_{p_{\theta_1}}[\langle v, T(X)\rangle] - \mathbb{E}_{p_{\theta_2}}[\langle v, T(X)\rangle]$$

$$= \sup_{\|v\|_2 \le 1}\langle v, \nabla A(\theta_1) - \nabla A(\theta_2)\rangle = \|\nabla A(\theta_1) - \nabla A(\theta_2)\|_2.$$

By the assumption on $\nabla^2 A$ we have that

$$\gamma\|\theta_1 - \theta_2\|_2 \le W_{\mathcal{F}}(p_{\theta_1}, p_{\theta_2}) \le \beta\|\theta_1 - \theta_2\|_2 \tag{17}$$

Moreover, the exponential family also satisfies that

$$D_{\mathrm{kl}}(p_{\theta_1}\|p_{\theta_2}) = A(\theta_2) - A(\theta_1) - \langle\nabla A(\theta_1), \theta_2 - \theta_1\rangle = \int_0^1 \rho^\top\nabla^2 A(\theta_2 + t\rho)\rho\,dt$$

where $\rho = \theta_1 - \theta_2$. Using the assumption we have that $\gamma\|\theta_1 - \theta_2\|^2 \le \rho^\top\nabla^2 A(\theta_2 + t\rho)\rho \le \beta\|\theta_1 - \theta_2\|^2$ and therefore $\frac{1}{2}\gamma\|\theta_1 - \theta_2\|^2 \le D_{\mathrm{kl}}(p_{\theta_1}\|p_{\theta_2}) \le \frac{1}{2}\beta\|\theta_1 - \theta_2\|^2$. Combining this with eq. (17) we complete the proof.

**Wasserstein bounds** We show eq. (8). As $\mathrm{diam}(\mathcal{X}) = D$, there exists $x_0 \in \mathcal{X}$ such that $\|x - x_0\| \leq D$ for all $x \in \mathcal{X}$. Hence for any 1-Lipschitz function $f : \mathbb{R}^d \to \mathbb{R}$ we have that $|f(X) - f(x_0)| \leq \|x - x_0\|_2 \leq D$. By the Hoeffding Lemma, $f(X)$ is $D^2/4$-sub-Gaussian. Applying Theorem D.1(a), we get that for any $p, q \in \mathcal{G}$,

$$W_1(p, q) \leq \sqrt{\frac{D^2}{2} D_{\mathrm{kl}}(p\|q)} \lesssim \frac{D\gamma}{\sqrt{\beta}} \cdot W_{\mathcal{F}}(p, q).$$

**Generalization** For any $\theta \in \Theta$ we compute the Rademacher complexity

$$R_n(\mathcal{F}, p_\theta) = \mathbb{E}_{p_\theta}\left[\sup_{\|v\|_2 \leq 1} \left|\frac{1}{n}\sum_{i=1}^n \varepsilon_i \langle v, T(X_i)\rangle\right|\right] = \mathbb{E}_{p_\theta}\left[\left\|\frac{1}{n}\sum_{i=1}^n \varepsilon_i T(X_i)\right\|_2\right]$$

$$\leq \sqrt{\mathbb{E}_{p_\theta}\left[\left\|\frac{1}{n}\sum_{i=1}^n \varepsilon_i T(X_i)\right\|_2^2\right]} = \sqrt{\frac{E_{p_\theta}[\|T(X)\|_2^2]}{n}}.$$

## C    RESULTS ON MIXTURE OF GAUSSIANS

We consider mixture of $k$ identity-covariance Gaussians on $\mathbb{R}^d$:

$$p_\theta = \sum_{i=1}^k w_i \mathsf{N}(\mu_i, I_d), \quad w_i \geq 0, \ \sum_{i=1}^k w_i = 1.$$

We assume that $\theta \in \Theta = \{\|\mu_i\|_2 \leq D, w_i \geq \exp(-B_w) \ i \in [k]\}$.

We will use a one-hidden-layer neural network that implements (a slight modification of) $\log p_\theta$:

$$\mathcal{F} = \left\{ f_1 - f_2 : f_i = \log \sum_{j=1}^k w_j^{(i)} \exp\left(\mu_j^{(i)\top} x + b_j^{(i)}\right) : \exp(-B_w) \leq w_j^{(i)} \leq 1, \ \left\|\mu_j^{(i)}\right\|_2 \leq D, \ 0 \geq b_j^{(i)} \geq -D^2 \right\}.$$

**Theorem C.1.** *The family $\mathcal{F}$ is suitable for learning mixture of $k$ Gaussians. Namely, we have that*

*(1) (Restricted approximability) For any $\theta_1, \theta_2 \in \Theta$, we have*

$$\frac{1}{D^2 + 1} \cdot W_1^2(p_{\theta_1}, p_{\theta_2}) \leq W_{\mathcal{F}}(p_{\theta_1}, p_{\theta_2}) \leq 2D \cdot W_1(p_{\theta_1}, p_{\theta_2}).$$

*(2) (Generalization) We have for some absolute constant $C > 0$ that*

$$\sup_{\theta \in \Theta} R_n(\mathcal{F}, p_\theta) \leq C\sqrt{\frac{k(\log k + D^2 + B_w)d\log n}{n}}.$$

### C.1    THE GAUSSIAN CONCENTRATION RESULT

The Gaussian concentration result (Vershynin, 2010, Proposition 5.34) will be used here and in later proofs, which we provide for convenience.

**Lemma C.2** (Gaussian concentration). *Suppose $X \sim \mathsf{N}(0, I_d)$ and $f : \mathbb{R}^d \to \mathbb{R}$ is L-Lipschitz, then $f(X)$ is $L^2$-sub-Gaussian.*

### C.2    PROOF OF THEOREM C.1

**Restricted approximability** For the upper bound, it suffices to show that each

$$f(x) = \log \sum_{j=1}^k w_j \exp\left(\mu_j^\top x + b_j\right) \tag{18}$$

is $D$-Lipschitz. Indeed, we have

$$\|\nabla \log f(x)\|_2 = \left\| \frac{\sum_{j=1}^k w_j \exp(\mu_j^\top x + b_j) \mu_j}{\sum_{j=1}^k w_j \exp(\mu_j^\top x + b_j)} \right\|_2 \leq \frac{\sum_{j=1}^k w_j \exp(\mu_j^\top x + b_j) \|\mu_j\|_2}{\sum_{j=1}^k w_j \exp(\mu_j^\top x + b_j)} \leq D.$$

This further shows that every discriminator $f_1 - f_2 \in \mathcal{F}$ is at most $2D$-Lipschitz, so by Theorem D.2(a) we get the upper bound.

We now establish the lower bound. As $\mathcal{F}$ implements the KL divergence, for any two $p_1, p_2 \in \mathcal{P}$, we have

$$W_{\mathcal{F}}(p_1, p_2) \geq D_{\mathrm{kl}}(p_1\|p_2) + D_{\mathrm{kl}}(p_2\|p_1).$$

We consider regularity properties of the distributions $p_1, p_2$ in the Bobkov-Gotze sense (Theorem D.1(a)). Suppose $p_1 = \sum w_j \mathsf{N}(\mu_j, I_d)$. For any 1-Lipschitz function $f : \mathbb{R}^d \to \mathbb{R}$, we have

$$f(X) \stackrel{d}{=} \sum_{j=1}^j f(\mathsf{N}(\mu_j, I_d)).$$

Letting $X_j \sim \mathsf{N}(\mu_j, I_d)$ be the mixture components. By the Gaussian concentration (Lemma C.2), each $f(X_j)$ is 1-sub-Gaussian, so we have for any $\lambda \in \mathbb{R}$

$$\mathbb{E}[e^{\lambda f(X)}] = \sum_{j=1}^k w_j \mathbb{E}[e^{\lambda f(X_j)}] \leq \sum_{j=1}^k e^{\lambda \mathbb{E}[f(X_j)] + \lambda^2/2} = e^{\lambda^2/2} \underbrace{\sum_{j=1}^k w_j e^{\lambda \mathbb{E}[f(X_j)]}}_{\mathrm{I}}.$$

Now, term I is precisely the MGF of a discrete random variable on $Y \in \mathbb{R}$ which takes value $\mathbb{E}[f(X_j)]$ with probability $w_j$. For $Z \sim \mathsf{N}(0, 1)$ we have

$$|\mathbb{E}[f(X_j)] - \mathbb{E}[f(Z)]| = |\mathbb{E}[f(\mu_j + Z)] - \mathbb{E}[f(Z)]| \leq \mathbb{E}[|f(\mu_j + Z) - f(Z)|] \leq \|\mu_j\|_2 \leq D.$$

Therefore the values $\{\mathbb{E}[f(X_j)]\}_{j \in [k]}$ lie in an interval of length at most $2D$. By the Hoeffding's Lemma, $Y$ is $D^2$-sub-Gaussian, so we have $\mathrm{I} \leq \exp(\lambda \mathbb{E}[Y] + D^2\lambda^2/2)$, and so

$$\mathbb{E}[e^{\lambda f(X)}] \leq \exp\left(\frac{\lambda^2}{2} + \lambda \mathbb{E}[Y] + D^2\lambda^2/2\right) = \exp\left(\lambda \mathbb{E}[Y] + \frac{\lambda^2(D^2 + 1)}{2}\right).$$

Therefore $f(X)$ is at most $(D^2+1)$-sub-Gaussian, and thus $X$ satisfies the Bobkov-Gozlan condition with $\sigma^2 = D^2 + 1$. Applying Theorem D.1(a) we get

$$W_{\mathcal{F}}(p_1, p_2) \geq D_{\mathrm{kl}}(p_1\|p_2) + D_{\mathrm{kl}}(p_2\|p_1) \geq \frac{1}{D^2 + 1} \cdot W(p_1, p_2).$$

**Generalization**   Reparametrize the one-hidden-layer neural net eq. (18) as

$$f_\theta(x) = \log \sum_{j=1}^k \exp(\mu_j^\top x + \underbrace{b_j + \log w_j}_{c_j}).$$

It then suffices to bound the Rademacher complexity of $f_\theta$ for $\theta \in \Theta = \left\{\|\mu_j\|_2 \leq D, c_j \in [-(D^2 + B_w), 0]\right\}$. Define the metric

$$\rho(\theta, \theta') = \max_{j \in [k]} \max \left\{\|\mu_j - \mu_j'\|_2, |c_j - c_j'|\right\}$$

and the Rademacher process

$$Y_\theta = \frac{1}{n} \sum_{i=1}^n \varepsilon_i f_\theta(X_i) = \frac{1}{n} \sum_{i=1}^n \varepsilon_i \log \sum_{j=1}^k \exp(\mu_j^\top X_i + c_j),$$

we show that $Y_\theta$ is suitably Lipschitz in $\theta$ (in the $\rho$ metric) and use a one-step discretization bound. Indeed, we have

$$\|\nabla_{\mu_j} Y_\theta\|_2 = \left\| \frac{1}{n} \sum_{i=1}^n \varepsilon_i \frac{\exp(\mu_j^\top X_i + c_j)}{\sum_{j=1}^k \exp(\mu_j^\top X_i + c_j)} X_i \right\|_2 \leq \frac{1}{n} \sum_{i=1}^n \|X_i\|_2$$

and

$$\left| \nabla_{c_j} Y_\theta \right| = \left| \frac{1}{n} \sum_{i=1}^n \varepsilon_i \frac{\exp(\mu_j^\top X_i + c_j)}{\sum_{j=1}^k \exp(\mu_j^\top X_i + c_j)} \right| \le 1.$$

Therefore, for any $\varepsilon > 0$ we have

$$\mathbb{E}\left[ \sup_{\rho(\theta,\theta') \le \varepsilon} |Y_\theta - Y_{\theta'}| \right] \le Ck \left( \mathbb{E}\left[ \|X_1\|_2 \right] + 1 \right) \varepsilon \le Ck(D + \sqrt{d})\varepsilon \tag{19}$$

for some constant $C > 0$.

We now bound the expected supremum of the max over a covering set. Let $\mathcal{N}(\Theta, \rho, \varepsilon)$ be a $\varepsilon$-covering set of $\Theta$ under $\rho$, and $N(\Theta, \rho, \varepsilon)$ be the covering number. As $\rho$ looks at each $\mu_i, c_j$ separately, its covering number can be upper bounded by the product of each separate covering:

$$N(\Theta, \rho, \varepsilon) \le \prod_{j=1}^k N(\mathsf{B}_2(D), \|\cdot\|_2, \varepsilon) \cdot N([-(D^2+B_w), 0], |\cdot|, \varepsilon) \le \exp\left( kd \log \frac{3D}{\varepsilon} + k \log \frac{2(D^2 + B_w)}{\varepsilon} \right).$$

Now, for each invididual process $Y_\theta$ is the i.i.d. average of random variables of the form $\varepsilon_i \log \sum_{j=1}^k \exp(\mu_j^\top X + c_j)$. The log-sum-exp part is $D$-Lipschitz in $X$, so we can reuse the analysis done precedingly (in the Bobkov-Gotze part) to get that $\log \sum_{j=1}^k \exp(\mu_j^\top X + c_j)$ is $D^2(D^2 + 1)$-sub-Gaussian. Further, its expectation is bounded as (for $X \sim p = \sum v_i \mathsf{N}(\nu_i, I_d)$)

$$\mathbb{E}_{X \sim p}\left[ \log \sum_{j=1}^k \exp(\mu_j^\top X + c_j) \right] = \sum_{i=1}^k v_i \mathbb{E}_{X \sim \mathsf{N}(\nu_i, I_d)}\left[ \log \sum_{j=1}^k \exp(\mu_j^\top X + c_j) \right]$$

$$\le \sum_{i=1}^k v_i \log \sum_{j=1}^k \mathbb{E}_{X \sim \mathsf{N}(\nu_i, I_d)}[\exp(\mu_j^\top X + c_j)] \le \sum_{i=1}^k v_i \log \sum_{j=1}^k \exp(\mu_j^\top \nu_i + \|\mu_j\|_2^2 / 2 + c_j)$$

$$\le \log k + (2D^2 + B_w).$$

This shows that the term $\varepsilon_i \log \sum_{j=1}^k \exp(\mu_j^\top X + c_j)$ is $(\log k + D^2 + B_w)^2 + D^2(D^2 + 1)$-sub-Gaussian, and thus we have by sub-Gaussian maxima bounds that

$$\mathbb{E}\left[ \max_{\theta \in \mathcal{N}(\Theta,\rho,\varepsilon)} |Y_\theta| \right] \le C \sqrt{\frac{(\log k + D^2 + B_w)^2 + D^2(D^2 + 1)}{n} \cdot \log N(\Theta, \rho, \varepsilon)}$$

$$\le C \sqrt{\frac{\log k + D^2 + B_w}{n} \cdot kd \log \frac{D^2 + B_w}{\varepsilon}}. \tag{20}$$

By the 1-step discretization bound and combining eq. (19) and appendix C.2, we get

$$\mathbb{E}\left[ \sup_{\theta \in \Theta} |Y_\theta| \right] \le \mathbb{E}\left[ \sup_{\theta,\theta' \in \Theta, \rho(\theta,\theta') \le \varepsilon} |Y_\theta - Y_{\theta'}| \right] + \mathbb{E}\left[ \max_{\theta \in \mathcal{N}(\Theta,\rho,\varepsilon)} |Y_\theta| \right]$$

$$\le Ck(D + \sqrt{d})\varepsilon + C \sqrt{\frac{\log k + D^2 + B_w}{n} \cdot kd \log \frac{D^2 + B_w}{\varepsilon}}.$$

Choosing $\varepsilon = c/n$ for sufficiently small $c$ (depending on $D^2, B_w$) gives that

$$\mathbb{E}\left[ \sup_{\theta \in \Theta} |Y_\theta| \right] \le C \sqrt{\frac{kd(\log k + D^2 + B_w) \log n}{n}}$$

## D    PROOFS FOR SECTION 4

### D.1    BOUNDING KL BY WASSERSTEIN

The following theorem gives conditions on which the KL divergence can be lower bounded by the Wasserstein 1/2 distance. For a reference see Section 4.1 and 4.4 in van Handel (2014).

**Theorem D.1** (Lower bound KL by Wasserstein). *Let $p$ be any distribution on $\mathbb{R}^d$ and $X_i \overset{\text{iid}}{\sim} p$ be the i.i.d. samples from $p$.*

*(a) (Bobkov-Gotze) If $f(X_1)$ is $\sigma^2$-sub-Gaussian for any 1-Lipschitz $f : \mathbb{R}^d \to \mathbb{R}$, then*
$$W_1(p,q)^2 \leq 2\sigma^2 D_{\mathrm{kl}}(p\|q) \quad \text{for all } q.$$

*(b) (Gozlan) If $f(X_1, \ldots, X_n)$ is $\sigma^2$-sub-Gaussian for any 1-Lipschitz $f : (\mathbb{R}^d)^n \to \mathbb{R}$, then*
$$W_2(p,q)^2 \leq 2\sigma^2 D_{\mathrm{kl}}(p\|q) \quad \text{for all } q.$$

**Theorem D.2** (Upper bounding $f$-contrast by Wasserstein). *Let $p, q$ be two distributions on $\mathbb{R}^d$ with positive densities and denote their probability measures by $P, Q$. Let $f : \mathbb{R}^d \to \mathbb{R}$ be a function.*

*(a) ($W_1$ bound) Suppose $f$ is L-Lipschitz, then $\mathbb{E}_p[f(X)] - \mathbb{E}_q[f(X)] \leq L \cdot W_1(p,q)$.*

*(b) (Truncated $W_1$ bound) Let $D > 0$ be any diameter of interest. Suppose for any $\widetilde{p} \in \{p, q\}$ we have*

  *(i)  $f$ is $L(D)$-Lipschitz in the ball of radius $D$;*
  *(ii)  We have $\mathbb{E}_{\widetilde{p}}[f^2(X)] \leq M$;*
  *(iii)  We have $\widetilde{P}(\|X\| \geq D) \leq p_{\mathrm{tail}}(D)$,*

  *then we have*
$$\mathbb{E}_p[f(X)] - \mathbb{E}_q[f(X)] \leq L(D) \cdot W_1(p,q) + 4\sqrt{Mp_{\mathrm{tail}}(D)}.$$

*(c) ($W_2$ bound) Suppose $\|\nabla f(x)\| \leq c_1\|x\| + c_2$ for all $x \in \mathbb{R}^d$, then we have*
$$\mathbb{E}_p[f(X)] - \mathbb{E}_q[f(X)] \leq \left( \frac{c_1}{2}\sqrt{\mathbb{E}_p[\|X\|^2]} + \frac{c_1}{2}\sqrt{\mathbb{E}_q[\|X\|^2]} + c_2 \right) \cdot W_2(p,q).$$

### D.1.1   PROOF OF THEOREM D.2

*Proof.* (a) This follows from the dual formulation of $W_1$.

(b) We do a truncation argument. We have
$$\mathbb{E}_p[f(X)] - \mathbb{E}_q[f(X)] = \underbrace{\mathbb{E}_p[f(X)\mathbf{1}\{\|X\| \leq D\}] - \mathbb{E}_q[f(X)\mathbf{1}\{\|X\| \leq D\}]}_{\mathrm{I}}$$
$$+ \underbrace{\mathbb{E}_p[f(X)\mathbf{1}\{\|X\| > D\}] - \mathbb{E}_q[f(X)\mathbf{1}\{\|X\| > D\}]}_{\mathrm{II}}.$$

Term II has the followng bound by Cauchy-Schwarz:
$$\mathrm{II} \leq \sqrt{\mathbb{E}_p[f^2(X)] \cdot P(\|X\| > D)} + \sqrt{\mathbb{E}_q[f^2(X)] \cdot Q(\|X\| > D)} \leq 2\sqrt{Mp_{\mathrm{tail}}(D)}.$$

We now deal with term I. By definition of the Wasserstein distance, there exists a coupling $(X, Y) \sim \pi$ such that $X \sim P$, $Y \sim Q$, and $\mathbb{E}_\pi[\|X - Y\|] = W_1(p,q)$. On this coupling, we have

$$
\begin{aligned}
\mathrm{I} &= \mathbb{E}_\pi[f(X)\mathbf{1}\{\|X\| \leq D\} - f(Y)\mathbf{1}\{\|Y\| \leq D\}] \\
&= \mathbb{E}_\pi[(f(X) - f(Y))\mathbf{1}\{\|X\| \leq D, \|Y\| \leq D\}] \\
&\quad + \mathbb{E}_\pi[f(X)\mathbf{1}\{\|X\| \leq D, \|Y\| > D\}] - \mathbb{E}_\pi[f(Y)\mathbf{1}\{\|Y\| \leq D, \|X\| > D\}] \\
&\overset{(i)}{\leq} L(D)\mathbb{E}_\pi[\|X - Y\|\mathbf{1}\{\|X\| \leq D, \|Y\| \leq D\}] + \mathbb{E}_\pi[|f(X)|\mathbf{1}\{\|Y\| \geq D\}] \\
&\quad + \mathbb{E}_\pi[|f(Y)|\mathbf{1}\{\|X\| \geq D\}] \\
&\overset{(ii)}{\leq} L(D)\mathbb{E}_\pi[\|X - Y\|] + \sqrt{\mathbb{E}_\pi[f^2(X)] \cdot \pi(\|Y\| \geq D)} \\
&\quad + \sqrt{\mathbb{E}_\pi[f^2(Y)] \cdot \pi(\|X\| \geq D)} \\
&= L(D) \cdot W_1(p,q) + \sqrt{\mathbb{E}_p[f^2(X)] \cdot Q(\|X\| \geq D)} + \sqrt{\mathbb{E}_q[f^2(X)] \cdot P(\|X\| \geq D)} \\
&\leq L(D) \cdot W_1(p,q) + 2\sqrt{Mp_{\mathrm{tail}}(D)}.
\end{aligned}
$$

Above, inequality (i) used the Lipschitzness of $f$ in the $D$-ball, and (ii) used Cauchy-Schwarz. Putting terms I and II together we get

$$\mathbb{E}_p[f(X)] - \mathbb{E}_q[f(X)] \leq L(D) \cdot W_1(p,q) + 4\sqrt{Mp_{\text{tail}}(D)}.$$

(c) This part is a straightforward extension of (Polyanskiy & Wu, 2016, Proposition 1). For completeness we present the proof here. For any $x, y \in \mathbb{R}^d$ we have

$$|f(x) - f(y)| = \left| \int_0^1 \langle \nabla f(tx + (1-t)y), x - y \rangle \, dt \right| \leq \int_0^1 \|\nabla f(tx + (1-t)y)\| \, \|x - y\| \, dt$$

$$\leq \int_0^1 (c_1 t \|x\| + c_1(1-t)\|y\| + c_2) \|x - y\| \, dt = (c_2 + c_1 \|x\|/2 + c_1 \|y\|/2) \|x - y\|.$$

By definition of the $W_2$ distance, there exists a coupling $(X, Y) \sim \pi$ such that $X \sim P, Y \sim Q$, and $\mathbb{E}[\|X - Y\|^2] = W_2^2(p,q)$. On this coupling, taking expectation of the above bound, we get

$$\mathbb{E}_\pi[|f(X) - f(Y)|] \leq c_2 \mathbb{E}_\pi[\|X - Y\|] + \frac{c_1}{2} \left( \mathbb{E}_\pi[\|X\| \, \|X - Y\|] + \mathbb{E}_\pi[\|Y\| \, \|X - Y\|] \right)$$

$$\leq c_2 \sqrt{\mathbb{E}_\pi[\|X - Y\|^2]} + \frac{c_1}{2} \left( \sqrt{\mathbb{E}_\pi[\|X\|^2] \cdot \mathbb{E}_\pi[\|X - Y\|^2]} + \sqrt{\mathbb{E}_\pi[\|Y\|^2] \cdot \mathbb{E}_\pi[\|X - Y\|^2]} \right)$$

$$= \left( c_2 + \frac{c_1}{2} \sqrt{\mathbb{E}_p[\|X\|^2]} + \frac{c_1}{2} \sqrt{\mathbb{E}_q[\|X\|^2]} \right) \cdot W_2(p,q).$$

Finally, the triangle inequality gives

$$\mathbb{E}_p[f(X)] - \mathbb{E}_q[f(X)] = \mathbb{E}_\pi[f(X) - f(Y)] \leq \mathbb{E}_\pi[|f(X) - f(Y)|],$$

so the left hand side is also bounded by the preceding quantity.

$\square$

## D.2 PROOF OF LEMMA 4.1

It is straightforward to see that the inverse of $x = G_\theta(z)$ can be computed as

$$z = W_1^{-1}(\sigma^{-1}(W_2^{-1}\sigma^{-1}(\cdots \sigma^{-1}(W_\ell^{-1}(x - b_\ell)) \cdots) - b_2) - b_1). \tag{21}$$

So $G_\theta^{-1}$ is also a $\ell$-layer feedforward net with activation $\sigma^{-1}$.

We now consider the problem of representing $\log p_\theta(x)$ by a neural network. Let $\phi_\gamma$ be the density of $Z \sim \mathsf{N}(0, \text{diag}(\gamma^2))$. Recall that the log density has the formula

$$p_\theta(x) = \log \phi_\gamma \left( (G_\theta^{-1}(x)) + \log \left| \det \frac{\partial G_\theta^{-1}(x)}{\partial x} \right| \right..$$

First consider the inverse network that implements $G_\theta^{-1}$. By eq. (21), this network has $\ell$ layers ($\ell - 1$ hidden layers), $d^2 + d$ parameters in each layer, and $\sigma^{-1}$ as the activation function. Now, as $\log \phi_\gamma$ has the form $\log \phi_\gamma(z) = a(\gamma) - \sum_i z_i^2/(2\gamma_i^2)$, we can add one more layer on top of $z$ with the square activation and the inner product with $-\gamma^{-2}/2$ to get this term.

Second, we show that by adding some branches upon this network, we can also compute the log determinant of the Jacobian. Define $h_\ell = W_\ell^{-1}(x - b_\ell)$ and backward recursively $h_{k-1} = W_{k-1}^{-1}(\sigma^{-1}(h_k) - b_{k-1})$ (so that $z = h_1$), we have

$$\frac{\partial G_\theta^{-1}(x)}{\partial x} = W_1^{-1}\text{diag}(\sigma^{-1\prime}(h_2))W_2^{-1} \cdots W_{\ell-1}^{-1}\text{diag}(\sigma^{-1\prime}(h_\ell))W_\ell^{-1}.$$

Taking the log determinant gives

$$\log \left| \det \frac{\partial G_\theta^{-1}(x)}{\partial x} \right| = C + \sum_{k=2}^{\ell} \left\langle \mathbf{1}, \log \sigma^{-1\prime}(h_k) \right\rangle.$$

As $(h_\ell, \ldots, h_2)$ are exactly the (pre-activation) hidden layers of the inverse network, we can add one branch from each layer, pass it through the $\log \sigma^{-1'}$ activation, and take the inner product with $\mathbf{1}$.

Finally, by adding up the output of the density branch and the log determinant branch, we get a neural network that computes $\log p_\theta(x)$ with no more than $\ell + 1$ layers and $O(\ell d^2)$ parameters, and choice of activations within $\{\sigma^{-1}, \log \sigma^{-1'}, (\cdot)^2\}$.

### D.3   PROOF OF THEOREM 4.2

We state a similar restricted approximability bound here in terms of the $W_2$ distance, which we also prove.

$$W_2(p, q)^2 \lesssim D_{\mathrm{kl}}(p\|q) + D_{\mathrm{kl}}(q\|p) \leq W_\mathcal{F}(p, q) \lesssim \frac{\sqrt{d}}{\delta^2} \cdot W_2(p, q).$$

The theorem follows by combining the following three lemmas, which we show in sequel.

**Lemma D.3** (Lower bound). *There exists a constant $c = c(R_W, R_b, \ell) > 0$ such that for any $\theta_1, \theta_2 \in \Theta$, we have*

$$W_\mathcal{F}(p_{\theta_1}, p_{\theta_2}) \geq c \cdot W_2(p_{\theta_1}, p_{\theta_2})^2 \geq c \cdot W_1(p_{\theta_1}, p_{\theta_2})^2.$$

**Lemma D.4** (Upper bound). *There exists constants $C_i = C_i(R_W, R_b, \ell) > 0$, $i = 1, 2$ such that for any $\theta_1, \theta_2 \in \Theta$, we have*

*(1)  ($W_1$ bound) $W_\mathcal{F}(p_{\theta_1}, p_{\theta_2}) \leq \frac{C_1 \sqrt{d}}{\delta^2} \cdot \left(W_1(p_{\theta_1}, p_{\theta_2}) + \sqrt{d} \exp(-10d)\right)$.*

*(2)  ($W_2$ bound) $W_\mathcal{F}(p_{\theta_1}, p_{\theta_2}) \leq \frac{C_2 \sqrt{d}}{\delta^2} \cdot W_2(p_{\theta_1}, p_{\theta_2})$.*

**Lemma D.5** (Generalization error). *Consider $n$ samples $X_i \overset{\mathrm{iid}}{\sim} p_{\theta^\star}$ for some $\theta^\star \in \Theta$. There exists a constant $C = C(R_W, R_b, \ell) > 0$ such that when $n \geq C \max\left\{d, \delta^{-8} \log n\right\}$, we have*

$$R_n(\mathcal{F}, p_{\theta^\star}) \leq \sqrt{\frac{C d^4 \log n}{\delta^4 n}}.$$

### D.4   PROOF OF LEMMA D.3

We show that $p_\theta$ satisfies the Gozlan condition for any $\theta \in \Theta$ and apply Theorem D.1. Let $X_i \overset{\mathrm{iid}}{\sim} p_\theta$ for $i \in [n]$. By definition, we can write

$$X_i = G_\theta(Z_i), \quad Z_i \overset{\mathrm{iid}}{\sim} \mathsf{N}(0, \mathrm{diag}(\gamma^2)).$$

Let $\widetilde{Z}_i = (z_i/\gamma_i)_{i=1}^d$ and $\widetilde{G}_\theta(\widetilde{z}) = G_\theta((\gamma_i \widetilde{z}_i)_{i=1}^d)$, then we have $G_\theta(z) = \widetilde{G}_\theta(\widetilde{z})$ and that $\widetilde{Z}_i$ are i.i.d. standard Gaussian. Further, suppose $G_\theta$ is $L$-Lipschitz, then for all $z_1, z_2 \in \mathbb{R}^d$ we have

$$\left|\widetilde{G}_\theta(\widetilde{z}_1) - \widetilde{G}_\theta(\widetilde{z}_2)\right| = |G_\theta(z_1) - G_\theta(z_2)| \leq L \|z_1 - z_2\|_2 \leq L \|\widetilde{z}_1 - \widetilde{z}_2\|_2,$$

the last inequality following from $\gamma_i \leq 1$. Therefore $\widetilde{G}_\theta$ is also $L$-Lipschitz.

Now, for any 1-Lipschitz $f : (\mathbb{R}^d)^n \to \mathbb{R}$, we have

$$\left|f(\widetilde{G}_\theta(\widetilde{z}_1), \ldots, \widetilde{G}_\theta(\widetilde{z}_n)) - f(\widetilde{G}_\theta(\widetilde{z}_1'), \ldots, \widetilde{G}_\theta(\widetilde{z}_n'))\right| \leq \left(\sum_{i=1}^n \left\|\widetilde{G}_\theta(\widetilde{z}_i) - \widetilde{G}_\theta(\widetilde{z}_i')\right\|_2^2\right)^{1/2}$$

$$\leq \quad L \cdot \left(\sum_{i=1}^n \|\widetilde{z}_i - \widetilde{z}_i'\|_2^2\right)^{1/2} = L \left\|\widetilde{z}_{1:n} - \widetilde{z}'_{1:n}\right\|_2.$$

Therefore the mapping $(\widetilde{z}_1, \ldots, \widetilde{z}_n) \to f(\widetilde{G}_\theta(\widetilde{z}_1), \ldots, \widetilde{G}_\theta(\widetilde{z}_n))$ is $L$-Lipschitz. Hence by Lemma C.2, the random variable

$$f(X_1, \ldots, X_n) = f(\widetilde{G}_\theta(\widetilde{Z}_1), \ldots, \widetilde{G}_\theta(\widetilde{Z}_n))$$

is $L^2$-sub-Gaussian, and thus the Gozlan condition is satisfied with $\sigma^2 = L^2$. By definition of the network $G_\theta$ we have

$$L \leq \prod_{k=1}^{\ell} \|W_k\|_{\mathrm{op}} \cdot \kappa_\sigma^{\ell-1} \leq {R_W}^\ell \kappa_\sigma^{\ell-1} = C(R_W, \ell).$$

Now, for any $\theta_1, \theta_2 \in \Theta$, we can apply Theorem D.1(b) and get

$$D_{\mathrm{kl}}(p_{\theta_1} \| p_{\theta_2}) \geq \frac{1}{2C^2} W_2(p_{\theta_1}, p_{\theta_2})^2,$$

and the same holds with $p_{\theta_1}$ and $p_{\theta_2}$ swapped. As $\log p_{\theta_1} - \log p_{\theta_2} \in \mathcal{F}$, by Lemma 4.3, we obtain

$$W_{\mathcal{F}}(p_{\theta_1}, p_{\theta_2}) \geq D_{\mathrm{kl}}(p_{\theta_1} \| p_{\theta_2}) + D_{\mathrm{kl}}(p_{\theta_2} \| p_{\theta_1}) \geq \frac{1}{C^2} W_2(p_{\theta_1}, p_{\theta_2})^2 \geq \frac{1}{C^2} W_1(p_{\theta_1}, p_{\theta_2})^2,$$

The last bound following from the fact that $W_2 \geq W_1$.

### D.5 PROOF OF LEMMA D.4

We are going to upper bound $W_{\mathcal{F}}$ by the Wasserstein distances through Theorem D.2. Fix $\theta_1, \theta_2 \in \Theta$. By definition of $\mathcal{F}$, it suffices to upper bound the Lipschitzness of $\log p_\theta(x)$ for all $\theta \in \Theta$. Recall that

$$\log p_\theta(x) = \underbrace{\frac{1}{2} \left\langle h_\ell, \mathrm{diag}(\gamma^{-2}) h_\ell \right\rangle}_{\mathrm{I}} + \underbrace{\sum_{k=1}^{\ell-1} \left\langle \mathbf{1}_d, \log \sigma^{-1'}(h_k) \right\rangle}_{\mathrm{II}} + C(\theta),$$

where $h_1, \ldots, h_\ell(= z)$ are the hidden-layers of the inverse network $z = G_\theta^{-1}(x)$, and $C(\theta)$ is a constant that does not depend on $x$.

We first show the $W_2$ bound. Clearly $\log p_\theta(x)$ is differentiable in $x$. As $\theta \in \Theta$ has norm bounds, each layer $h_k$ is $C(R_W, R_b, k)$-Lipschitz in $x$, so term II is altogether $\sqrt{d} \beta_\sigma \sum_{k=1}^{\ell-1} C(R_W, R_b, k) = C(R_W, R_b, \ell) \sqrt{d}$-Lipschitz in $x$. For term I, note that $h_\ell$ is $C$-Lipschitz in $x$, so we have

$$\|\nabla_x \mathrm{I}\|_2 \leq \frac{1}{\min_i \gamma_i^2} \|h_\ell(x)\|_2 \|\nabla_x h_\ell(x)\|_2 \leq \frac{1}{\delta^2} (C \|x\|_2 + h_\ell(0)) \cdot C \leq \frac{C}{\delta^2} (1 + \|x\|_2).$$

Putting together the two terms gives

$$\|\nabla \log p_\theta(x)\|_2 \leq \frac{C}{\delta^2} (1 + \|x\|_2) + \frac{C}{\delta^2} \sqrt{d} \leq \frac{C}{\delta^2} (\|x\|_2 + \sqrt{d}). \tag{22}$$

Further, under either $p_{\theta_1}$ or $p_{\theta_2}$ (for example $p_{\theta_1}$), we have

$$\mathbb{E}_{p_{\theta_1}}[\|X\|_2^2] \leq \mathbb{E}[\|G_{\theta_1}(Z)\|_2^2] \leq C(R_W, R_b, \ell) \mathbb{E}[(\|Z\|_2 + 1)^2] \leq Cd.$$

Therefore we can apply Theorem D.2(c) and get

$$\mathbb{E}_{p_{\theta_1}}[\log p_\theta(x)] - \mathbb{E}_{p_{\theta_2}}[\log p_\theta(x)] \leq \frac{C}{\delta^2} \left(2\sqrt{Cd} + \sqrt{d}\right) W_2(p_{\theta_1}, p_{\theta_2}) \leq \frac{C\sqrt{d}}{\delta^2} W_2(p_{\theta_1}, p_{\theta_2}).$$

We now turn to the $W_1$ bound. The bound eq. (22) already implies that for $\|X\|_2 \leq D$,

$$\|\nabla \log p_\theta(x)\|_2 \leq \frac{C}{\delta^2} (D + \sqrt{d}).$$

Choosing $D = K\sqrt{d}$, for a sufficiently large constant $K$, by the bound $\|X\|_2 \leq C(\|Z\|_2 + 1)$ we have the tail bound

$$\mathbb{P}(\|X\|_2 \geq D) \leq \exp(-20d).$$

On the other hand by the bound $|\log p_\theta(x)| \leq C((\|x\|_2 + 1)^2/\delta^2 + \sqrt{d}(\|x\|_2 + 1))$ we get under either $p_{\theta_1}$ or $p_{\theta_2}$ (for example $p_{\theta_1}$) we have

$$\mathbb{E}_{p_{\theta_1}} \left[(\log p_\theta(X))^2\right] \leq \frac{C}{\delta^4} \mathbb{E}\left[\left(\|X\|_2^2 + \sqrt{d}(\|X\|_2 + 1)\right)^2\right] \leq \frac{Cd^2}{\delta^4}.$$

Thus we can substitute $D = K\sqrt{d}$, $L(D) = C(1 + K)\sqrt{d}/\delta^2$, $M = Cd^2/\delta^4$, and $p_{\text{tail}}(D) = \exp(-2d)$ into Theorem D.2(b) and get

$$\mathbb{E}_{p_{\theta_1}}[\log p_\theta(x)] - \mathbb{E}_{p_{\theta_2}}[\log p_\theta(x)] \leq \frac{C(1+K)\sqrt{d}}{\delta^2} W_1(p_{\theta_1}, p_{\theta_2}) + 4\sqrt{\frac{Cd^2}{\delta^4} \exp(-20d)}$$

$$\leq \frac{C\sqrt{d}}{\delta^2} \left( W_1(p_{\theta_1}, p_{\theta_2}) + \sqrt{d}\exp(-10d) \right).$$

### D.6    PROOF OF LEMMA D.5

For any log-density neural network $F_\theta(x) = \log p_\theta(x)$, reparametrize so that $(W_i, b_i)$ represent the weights and the biases of the inverse network $z = G_\theta^{-1}(x)$. By eq. (21), this has the form

$$(W_i, b_i) \longleftarrow (W_{\ell-i+1}^{-1}, -W_{\ell-i+1}^{-1} b_{\ell-i+1}), \ \ \forall i \in [\ell].$$

Consequently the reparametrized $\theta = (W_i, b_i)_{i\in[\ell]}$ belongs to the (overloading $\Theta$)

$$\Theta = \left\{ \theta = (W_i, b_i)_{i=1}^\ell : \ \max\left\{ \|W_i\|_{\text{op}}, \|W_i^{-1}\|_{\text{op}} \right\} \leq R_W, \ \|b_i\|_2 \leq R_W R_b, \ \forall i \in [\ell] \right\}. \quad (23)$$

As $\mathcal{F} = \{F_{\theta_1} - F_{\theta_2} : \theta_1, \theta_2 \in \Theta\}$, the Rademacher complexity of $\mathcal{F}$ is at most two times the quantity

$$R_n := R_n(\{F_\theta : \theta \in \Theta\}, p_{\theta^\star}) = \left[ \sup_{\theta \in \Theta} \left| \frac{1}{n} \sum_{i=1}^n \varepsilon_i F_\theta(X_i) \right| \right],$$

We do one additional re-parametrization. Note that the log-density network $F_\theta(x) = \log p_\theta(x)$ has the form

$$F_\theta(x) = \phi_\gamma(G_\theta^{-1}(x)) + \log\left| \det \frac{\partial G_\theta^{-1}(x)}{\partial x} \right| + K(\theta) = \frac{1}{2} \langle h_\ell, \text{diag}(\gamma^{-2})h_\ell \rangle + \log\left| \det \frac{\partial G_\theta^{-1}(x)}{\partial x} \right| + K(\theta). \quad (24)$$

The constant $C(\theta)$ is the sum of the normalizing constant for Gaussian density (which is the same across all $\theta$, and as we are taking subtractions of two $\log p_\theta$, we can ignore this) and the sum of $\log\det(W_i)$, which is upper bounded by $d\ell R_W$. We can additionally create a parameter $K = K(\theta) \in [0, d\ell R_W]$ for this term and let $\theta \leftarrow (\theta, K)$.

For any (reparametrized) $\theta, \theta' \in \Theta$, define the metric

$$\rho(\theta, \theta') = \max\left\{ \|W_i - W_i'\|_{\text{op}}, \|b_i - b_i'\|_2, |K - K'| : i \in [d] \right\}.$$

Then we have, letting $Y_\theta = \frac{1}{n} \sum_{i=1}^n \varepsilon_i F_\theta(X_i)$ denote the Rademacher process, the one-step discretization bound (Wainwright, 2018, Section 5).

$$R_n \leq \mathbb{E}\left[ \sup_{\theta, \theta' \in \Theta, \rho(\theta, \theta') \leq \varepsilon} |Y_\theta - Y_{\theta'}| \right] + \mathbb{E}\left[ \sup_{\theta_i \in \mathcal{N}(\Theta, \rho, \varepsilon)} |Y_{\theta_i}| \right]. \quad (25)$$

We deal with the two terms separately in the following two lemmas.

**Lemma D.6** (Discretization error). *There exists a constant $C = C(R_W, R_b, \ell)$ such that, for all $\theta, \theta' \in \Theta$ such that $\rho(\theta, \theta') \leq \varepsilon$, we have*

$$|Y_\theta - Y_{\theta'}| \leq \frac{C}{\delta^2} \frac{1}{n} \sum_{i=1}^n \left( \sqrt{d}(1 + \|X_i\|_2) + \|X_i\|_2^2 \right) \cdot \varepsilon.$$

**Lemma D.7** (Expected max over a finite set). *There exists constants $\lambda_0, C$ (depending on $R_W, R_b, \ell$ but not $d, \delta$) such that for all $\lambda \leq \lambda_0 \delta^2 n$,*

$$\mathbb{E}\left[ \max_{\theta_i \in \mathcal{N}(\Theta, \rho, \varepsilon)} |Y_\theta| \right] \leq \frac{Cd^2 \log \frac{\max\{R_W, R_b\}}{\varepsilon}}{\lambda} + \frac{Cd^2\lambda}{\delta^4 n}.$$

Substituting the above two Lemmas into the bound eq. (25), we get that for all $\varepsilon \le \min\{R_W, R_b\}$ and $\lambda \le \lambda_0 \delta^2 n$,

$$R_n \le \underbrace{\frac{C}{\delta^2}\mathbb{E}[\sqrt{d}(1 + \|X\|_2) + \|X\|_2^2] \cdot \varepsilon}_{\text{I}} + \underbrace{\frac{Cd^2 \log \frac{\max\{R_W, R_b\}}{\varepsilon}}{\lambda} + \frac{Cd^2\lambda}{\delta^4 n}}_{\text{II}}.$$

As $X = G_{\theta^\star}(Z)$, we have $\|X\|_2 \le C\|Z\|_2 + \|G_{\theta^\star}(0)\|_2 \le C(\|Z\|_2 + 1)$ for some constant $C > 0$. As $\mathbb{E}[\|Z\|_2^2] = k + \delta^2(d-k) \le d$, we have $\mathbb{E}[\sqrt{d}(1 + \|X\|_2) + \|X\|_2^2] \le Cd$ for some constant $C$, giving that I $\le \frac{Cd}{\delta^2} \cdot \varepsilon$. Choosing $\varepsilon = c/(dn)$ guarantees that I $\le \frac{1}{\delta^2 n}$. For this choice of $\varepsilon$, term II has the form

$$\text{II} \le \frac{Cd^2 \log(nd \max\{R_W, R_b\})}{\lambda} + \frac{Cd^2\lambda}{\delta^4 n} \le \frac{Cd^2 \log n}{\lambda} + \frac{Cd^2\lambda}{\delta^4 n},$$

the last bound holding if $n \ge d$. Choosing $\lambda = \sqrt{n \log n / \delta^4}$, which will be valid if $n/\log n \ge \delta^{-8}\lambda_0^{-2}$, we get

$$\text{II} \le Cd^2\sqrt{\frac{\log n}{\delta^4 n}} = C\sqrt{\frac{d^4 \log n}{\delta^4 n}}.$$

This term dominates term I and is hence the order of the generalization error.

### D.6.1 PROOF OF LEMMA D.6

Fix $\theta, \theta'$ such that $\rho(\theta, \theta') \le \varepsilon$. As $Y_\theta$ is the empirical average over $n$ samples and $|\varepsilon_i| \le 1$, it suffices to show that for any $x \in \mathbb{R}^d$,

$$|F_\theta(x) - F_{\theta'}(x)| \le \frac{C}{\delta^2}\left(\sqrt{d}(1 + \|x\|_2) + \|x\|_2^2\right) \cdot \varepsilon.$$

For the inverse network $G_\theta^{-1}(x)$, let $h_k(x) \in \mathbb{R}^d$ denote the $k$-th hidden layer:

$$h_1(x) = \sigma(W_1 x + b_1), \;\cdots, \; h_{\ell-1}(x) = \sigma(W_{\ell-1}h_{\ell-2}(x) + b_{\ell-1}), \; h_\ell(x) = W_\ell h_{\ell-1}(x) + b_\ell = G_\theta^{-1}(x).$$

Let $h_k'(x)$ denote the layers of $G_{\theta'}^{-1}(x)$ accordingly. Using this notation, we have

$$F_\theta(x) = \frac{1}{2}\left\langle h_\ell, \mathrm{diag}(\gamma^{-2})h_\ell \right\rangle + \sum_{k=1}^{\ell-1}\left\langle \mathbf{1}_d, \log \sigma^{-1'}(h_k) \right\rangle + K.$$

**Lipschitzness of hidden layers** We claim that for all $k$, we have

$$\|h_k\|_2 \le (R_W \kappa_\sigma)^k \|x\|_2 + R_b \sum_{j=1}^{k}(R_W \kappa_\sigma)^j, \tag{26}$$

and consequently when $\rho(\theta, \theta') \le \varepsilon$, we have

$$\|h_k - h_k'\|_2 \le C(R_W, R_b, k)\varepsilon(1 + \|x\|_2), \quad C(R_W, R_b, k) = O\left(\sum_{j=0}^{k-1} j(R_W \kappa_\sigma)^j(1 + R_b)\right). \tag{27}$$

We induct on $k$ to show these two bounds. For eq. (26), note that $h_0 = \|x\|_2$ and

$$\|h_k\|_2 = \|\sigma(W_k h_{k-1} + b_k)\|_2 \le \kappa_\sigma(R_W \|h_{k-1}\|_2 + R_W R_b),$$

so an induction on $k$ shows the bound. For eq. (27), note that

$$\|h_1 - h_1'\|_2 \le \|W_1 - W_1'\|_{\mathrm{op}} \|x\|_2 + \|b_1 - b_1'\|_2 \le \varepsilon(1 + \|x\|_2),$$

so the base case holds. Now, suppose the claim holds for the $(k-1)$-th layer, then for the $k$-th layer we have

$$
\begin{aligned}
\left\|h_k - h_k'\right\|_2 &= \left\|\sigma(W_k h_{k-1} + b_k) - \sigma(W_k h_{k-1}' + b_k')\right\|_2 \leq \kappa_\sigma \left(\left\|W_k h_{k-1} - W_k' h_{k-1}'\right\|_2 + \left\|b_k - b_k'\right\|_2\right) \\
&\leq \kappa_\sigma \left(\varepsilon + \|W_k\|_{\mathrm{op}} \left\|h_{k-1} - h_{k-1}'\right\|_2 + \left\|W_k - W_k'\right\|_{\mathrm{op}} \left\|h_{k-1}'\right\|_2\right) \\
&\leq \kappa_\sigma \left(\varepsilon + R_W C(R_W, R_b, k-1)\varepsilon(1 + \|x\|_2) + \varepsilon\left((R_W\kappa_\sigma)^{k-1}\|x\|_2 + R_b \sum_{j=1}^{k-1}(R_W\kappa_\sigma)^j\right)\right) \\
&\leq \varepsilon(1 + \|x\|_2) \underbrace{\left(\kappa_\sigma R_W C(R_W, R_b, k-1) + (1 + R_b)\sum_{j=1}^{k-1}(R_W\kappa_\sigma)^j\right)}_{C(R_W, R_b, k)},
\end{aligned}
$$

' verifying the result for layer $k$.

**Dealing with $(\cdot)^2$ and $\log\sigma^{-1'}$** For the $\log\sigma^{-1'}$ term, note that $|(\log\sigma^{-1'})'| = |\sigma^{-1''}/\sigma^{-1'}| \leq \beta_\sigma$ by assumption. So we have the Lipschitzness

$$
\left|\sum_{k=1}^{\ell-1}\left\langle \mathbf{1}_d, \log\sigma^{-1'}(h_k) - \log\sigma^{-1'}(h_k')\right\rangle\right| \leq \sqrt{d}\beta_\sigma \sum_{k=1}^{\ell-1}\|h_k - h_k'\|_2
$$

$$
\leq \sqrt{d}\,\beta_\sigma \underbrace{\sum_{k=1}^{\ell-1}C(R_W, R_b, k)}_{C} \cdot\varepsilon(1 + \|x\|_2) = C\sqrt{d}(1 + \|x\|_2)\cdot\varepsilon.
$$

For the quadratic term, let $A_\gamma = \mathrm{diag}(\gamma^{-2})$ for shorthand. Using the bound $(1/2)|\langle u, Au\rangle - \langle v, Av\rangle| \leq \|A\|_{\mathrm{op}}(\|v\|_2 \|u - v\|_2 + \|u - v\|_2^2/2)$, we get

$$
\frac{1}{2}|\langle h_\ell, A_\gamma h_\ell\rangle - \langle h_\ell', A_\gamma h_\ell'\rangle| \leq \|A_\gamma\|_{\mathrm{op}}\left(\|h_\ell\|_2 \|h_\ell - h_\ell'\|_2 + \|h_\ell - h_\ell'\|_2^2/2\right)
$$

$$
\leq \frac{1}{\delta^2}\left(C\cdot C(R_W, R_b, \ell)\varepsilon(1 + \|x\|_2)^2 + C(R_W, R_b, \ell)^2\varepsilon^2(1 + \|x\|_2)^2/2\right) \leq \frac{C}{\delta^2}(1 + \|x\|_2)^2\cdot\varepsilon.
$$

**Putting together** Combining the preceding two bounds and that $|K - K'| \leq \varepsilon$, we get

$$
|F_\theta(x) - F_{\theta'}(x)| \leq \left(\frac{C}{\delta^2}(1 + \|x\|_2)^2 + C\sqrt{d}(1 + \|x\|_2) + 1\right)\varepsilon \leq \frac{C}{\delta^2}(\sqrt{d}(1 + \|x\|_2) + \|x\|_2^2)\cdot\varepsilon.
$$

### D.6.2 PROOF OF LEMMA D.7

**Tail decay at a single $\theta$** Fixing any $\theta \in \Theta$, we show that the random variable

$$
Y_\theta = \frac{1}{n}\sum_{i=1}^n \varepsilon_i F_\theta(x_i) = \frac{1}{n}\sum_{i=1}^n \varepsilon_i\left(\langle h_\ell(x_i), A_\gamma h_\ell(x_i)\rangle + \sum_{k=1}^{\ell-1}\left\langle \mathbf{1}_d, \log\sigma^{-1'}(h_k(x_i))\right\rangle + K\right).
$$

is suitably sub-exponential. To do this, it suffices to look at a single $x$ and then use rules for independent sums.

First, each $\left\langle \mathbf{1}_d, \log\sigma^{-1}(h_k(x))\right\rangle$ is sub-Gaussian, with mean and sub-Gaussianity parameter $O(Cd)$. Indeed, we have

$$
\left\langle \mathbf{1}_d, \log\sigma^{-1}(h_k(x))\right\rangle = \left\langle \mathbf{1}_d, \log\sigma^{-1}(h_k(G_{\theta^\star}(z)))\right\rangle = \left\langle \mathbf{1}_d, \log\sigma^{-1}(h_k(\widetilde{G}_{\theta^\star}(\widetilde{z})))\right\rangle,
$$

where $\widetilde{z} = [z_{1:k}, z_{(k+1):d}/\delta]$ is standard Gaussian. Note that (1) Lipschitzness of $\widetilde{G}$ is bounded by that of $G$, which is some $C(R_W, R_b, \ell)$, (2) all hidden-layers are $C(R_W, R_b, \ell)$-Lipschitz (see eq. (27)), (3) $v \mapsto \sum_{j=1}^d \log\sigma^{-1}(v_j)$ is $\sqrt{d}\beta_\sigma$-Lipschitz. Hence the above term is a $C\sqrt{d}$-Lipschitz function of a standard Gaussian, so is $Cd$-sub-Gaussian by Gaussian concentration C.2. To bound the mean, use the bound

$$
\mathbb{E}\left|\left\langle \mathbf{1}_d, \log\sigma^{-1}(h_k(\widetilde{G}_{\theta^\star}(\widetilde{z})))\right\rangle\right| \leq \mathbb{E}[\sqrt{d}\beta_\sigma C\|\widetilde{z}\|_2] \leq Cd.
$$

As we have $\ell - 1$ terms of this form, their sum is still $Cd$-sub-Gaussian with a $O(Cd)$ mean (absorbing $\ell$ into $C$).

Second, the term $\langle h_\ell, A_\gamma h_\ell \rangle$ is a quadratic function of a sub-Gaussian random vector, hence is sub-exponential. Its mean is bounded by $\mathbb{E}[\|A_\gamma\|_{\mathrm{op}} \|h_\ell\|_2^2] \leq Cd/\delta^2$. Its sub-exponential parameter is $1/\delta^2$ times the sub-Gaussian parameter of $h_\ell$, hence also $Cd/\delta^2$. In particular, there exists a constant $\lambda_0 > 0$ such that for all $\lambda \leq \lambda_0 \delta^2$,

$$\mathbb{E}[\exp(\lambda \langle h_\ell, A_\gamma h_\ell \rangle)] \leq \exp\left( \frac{Cd}{\delta^2} \lambda + \frac{C^2 d^2 \lambda^2}{\delta^4} \right).$$

(See for example (Vershynin, 2010) for such results.) Also, the parameter $K$ is upper bounded by $d\ell R_W = Cd$.

Putting together, multiplying by $\varepsilon_i$ (which addes up the squared mean onto the sub-Gaussianity / sub-exponentiality and multiplies it by at most a constant) and summing over $n$, we get that $Y_\theta$ is mean-zero sub-exponential with the MGF bound

$$\mathbb{E}[\exp(\lambda Y_\theta)] = \left( \mathbb{E}\left[ \exp\left( \frac{\lambda}{n} \varepsilon_i \left( \langle h_\ell(x_i), A_\gamma h_\ell(x_i) \rangle + \sum_{k=1}^{\ell-1} \left\langle \mathbf{1}_d, \log \sigma^{-1'}(h_k(x_i)) \right\rangle \right) \right) \right] \right)^n$$

$$\leq \exp\left( \frac{Cd^2\lambda^2}{\delta^4 n} \right), \quad \forall \lambda \leq \lambda_0 \delta^2 n. \tag{28}$$

**Bounding the expected maximum** We use the standard covering argument to bound the expected maximum. Recall that $\rho(\theta, \theta') = \max\left\{ \|W_i - W_i'\|_{\mathrm{op}}, \|b_i - b_i'\|_2, |K - K'| \right\}$. Hence, the covering number of $\Theta$ is bounded by the product of independent covering numbers, which further by the volume argument is

$$N(\Theta, \rho, \varepsilon) \leq \prod_{k=1}^{\ell} \left( 1 + \frac{2R_W}{\varepsilon} \right)^{d^2} \cdot \prod_{k=1}^{\ell} \left( 1 + \frac{2R_b}{\varepsilon} \right)^{d} \cdot \left( 1 + \frac{CdR_W}{\varepsilon} \right)$$

$$\leq \exp\left( C \left( \ell d^2 \log \frac{3R_W}{\varepsilon} + \ell d \log \frac{3R_b}{\varepsilon} \right) \right).$$

Using Jensen's inequality and applying the bound appendix D.6.2, we get that for any $\lambda \leq \lambda_0 \delta^2 n$,

$$\mathbb{E}\left[ \max_{\theta_i \in \mathcal{N}(\Theta, \rho, \varepsilon)} Y_\theta \right] \leq \frac{1}{\lambda} \left( \log \mathbb{E}\left[ \sum_{i=1}^{N(\Theta, \rho, \varepsilon)} \exp(\lambda Y_{\theta_i}) \right] \right) \leq \frac{1}{\lambda} \left( \log N(\Theta, \rho, \varepsilon) + \frac{Cd^2\lambda^2}{\delta^4 n} \right)$$

$$\leq \frac{Cd^2 \log \frac{\max\{R_W, R_b\}}{\varepsilon}}{\lambda} + \frac{Cd^2\lambda}{\delta^4 n}.$$

# E  PROOFS FOR SECTION 4.2

## E.1  FORMAL THEOREM STATEMENT

Towards stating the theorem more quantitatively, we will need to specify a few quantities of the generator class that will be relevant for us.

First, for notational simplicity, we override the definition of $p_\theta^\beta$ by a truncated version of the convolution of $p$ and a Gaussian distribution. Concretely, let $D_z = \{z : \|z\| \leq \sqrt{d} \log^2 d, z \in \mathbb{R}^k\}$ be a truncated region in the latent space (which contains an overwhelming large part of the probability mass), and the let $D_x = \{x : \exists z \in D_z, \|G(z) - x\|_2 \leq \beta\sqrt{d} \log^2 d\}$ be the image of $D_z$ under $G_\theta$. Recall that

$$G_\theta(z) = \sigma\left( W_l \sigma\left( W_{l-1} \ldots \sigma\left( W_1 z + b_1 \right) + \ldots b_{l-1} \right) + b_l \right).$$

Then, let $p_\theta^\beta(x)$ be the distribution obtained by adding Gaussian noise with variance $\beta^2$ to a sample from $G_\theta$, and truncates the distribution to a very high-probability region (both in the latent variable

and observable domain.) Formally, let $p_\theta^\beta$ be a distribution over $\mathbb{R}^d$, s.t.

$$p_\theta^\beta(x) \propto \int_{z \in D_z} e^{-\|z\|^2} e^{-\frac{\|G_\theta(z)-x\|^2}{\beta^2}} dz, \forall x \in D_x \tag{29}$$

For notational convenience, denote by $f : \mathbb{R}^k \to \mathbb{R}$ the function $f(z) = -\|z\|^2 - \|G_\theta(z)-x\|^2/\beta^2$, and denote by $z^*$ a maximum of $f$.
Furthermore, whenever clear from the context, we will drop $\theta$ from $p_\theta$ and $G_\theta$.

We introduce several regularity conditions for the family of generators $\mathcal{G}$:

**Assumption E.1.** *We assume the following bounds on the partial derivatives of $f$: we denote $S :=$ $\max_{z \in D_z : \|z-z*\| \le \delta} \|\nabla^2(\|G_\theta(z)-x\|^2)\|$, and $\lambda_{\min} := \max_{z \in D_z : \|z-z*\| \le \delta} \lambda_{\min}(\nabla^2 \|G_\theta(z)-x\|^2)$.*
*Similarly, we denote $t(z) := k^3 \max_{|I|=3} \frac{\partial \|G_\theta(z)-x\|^2}{\partial I}(z)$. and $T = \max_{z:z \in D_z} |t(z)|$.*
*We will denote by $R$ an upper bound on the quantity $\frac{1}{L_\sigma^l} \prod_{j=i}^{L} \sigma_{\min}(W_j)$ and by $L_G$ an upper bound*
*on the quantity $L_G := 2^l L_\sigma^l \prod_{i=1}^{l} \frac{1}{\sigma_{\min}(W_i)}$.*
*Finally, we assume the inverse activation function is Lipschitz, namely $|\sigma^{-1}(x) - \sigma^{-1}(y)| \le L_\sigma|x - y|$.*

**Note on asymptotic notation:** For notational convenience, in this section, $\lesssim, \gtrsim$, as well as the Big-Oh notation will hide dependencies on $R, L_G, S, T$ (in the theorem statements we intentionally emphasize the polynomial dependencies on $d$.) The main theorem states that for certain $\mathcal{F}$, $\tilde{d}_\mathcal{F}$ approximates the Wasserstein distance.

**Theorem E.1.** *Suppose the generator class $\mathcal{G}$ satisfies the assumption E.1 and let $\mathcal{F}$ be the family of functions as defined in Theorem E.2. Then, we have that for every $p, q \in \mathcal{G}$,*

$$W_1(p,q) \lesssim \tilde{d}_\mathcal{F}(p,q) \lesssim poly(d) \cdot W_1(p,q)^{1/6} + \exp(-d). \tag{30}$$

*Furthermore, when $n \gtrsim poly(d)$ we have $R_n(\mathcal{F}, \mathcal{G}) \lesssim poly(d)\sqrt{\frac{\log n}{n}}$. Here $\lesssim$ hides dependencies on $R, L_G, S$, and $T$.*

The main ingredient in the proof will be the theorem that shows that there exists a parameterized family $\mathcal{F}$ that can approximate the log density of $p^\beta$ for every $p \in \mathcal{G}$.

**Theorem E.2.** *Let $\mathcal{G}$ satisfy the assumptions in Assumption E.1. For $\beta = O(poly(1/d))$, there exists a family of neural networks $\mathcal{F}$ of size $poly(\frac{1}{\beta}, d)$ such that for every distribution $p \in \mathcal{G}$, there exists $N \in \mathcal{F}$ satisfying:*

*(1) $N$ approximates $\log p$ for typical $x$: given input $x = G(z^*) + r$, for $\|r\| \le 10\beta\sqrt{d}\log d$, and $\|z^*\| \le 10\sqrt{d}\log d$ for $\beta = O(poly(1/d))$ it outputs $N(x)$, s.t.*

$$|N(x) - \log p^\beta(x)| = O_{poly(d)}(\beta\log(1/\beta)) + \exp(-d)$$

*(2) $N$ is globally an approximate lower bound of $p$: on any input $x$, $N$ outputs $N(x) \le \log p^\beta(x) + O_{poly(d)}(\beta\log(1/\beta)) + \exp(-d)$.*
*(3) $N$ approximates the entropy in the sense that: the output $N(x)$ satisfies $|\mathbb{E}_{p^\beta}N(x) - H(p^\beta)| = O_{poly(d)}(\beta\log(1/\beta)) + \exp(-d)$*
*Moreover, every function in $\mathcal{F}$ has Lipschitz constant $O(\frac{1}{\beta^4}poly(d))$.*

The approach will be as follows: we will approximate $p^\beta(x)$ essentially by a variant of Laplace's method of integration, using the fact that

$$p^\beta(x) = C \int_{z \in D_z} e^{-\|z\|^2 - \frac{\|G(z)-x\|^2}{\beta^2}} dz$$

for a normalization constant $C$ that can be calculated up to an exponentially small additive factor. When $x$ is typical (in case (1) of Theorem E.2), the integral will mostly be dominated by it's maximum value, which we will approximately calculate using a greedy "inversion" procedure.
When $x$ is a atypical, it turns out that the same procedure will give a lower bound as in (2).

We are ready to prove Theorem E.1, assuming the correctness of Theorem E.2:

*Proof of Theorem E.1.* By Theorem E.2, we have that there exist neural networks $N_1, N_2 \in \mathcal{F}$ that approximate $\log p^\beta$ and $\log q^\beta$ respectively in the sense of bullet (1)-(3) in Theorem E.2. Thus we have that by bullet (2) for distribution $q^\beta$, and bullet (3) for distribution $q^\beta$, we have

$$\mathbb{E}_{p^\beta}[N_1(x) - N_2(x)] \geq \mathbb{E}_{p^\beta}[\log p] - \mathbb{E}_{p^\beta}[\log q] - O(\beta \log 1/\beta) \tag{31}$$

Similarly, we have

$$\mathbb{E}_{q^\beta}[N_2(x) - N_1(x)] \geq \mathbb{E}_{q^\beta}[\log q] - \mathbb{E}_{q^\beta}[\log p] - O(\beta \log 1/\beta) \tag{32}$$

Combining the equations above, setting $f = N_1(x) - N_2(x)$, we obtain that

$$W_\mathcal{F}(p^\beta, q^\beta) \geq \mathbb{E}_{p^\beta}[f] - \mathbb{E}_{q^\beta}[f] \geq D_{\mathrm{kl}}(p^\beta\|q^\beta) + D_{\mathrm{kl}}(q^\beta\|p^\beta) - O(\beta \log 1/\beta) \tag{33}$$

Therefore, by definition, and Bobkov-Götze theorem ($D_{\mathrm{kl}}(p^\beta, q^\beta) \gtrsim W_1(p^\beta, q^\beta)$)

$$W_1(p, q) \leq W_1(p^\beta, q^\beta) + O(\beta) \lesssim (D_{\mathrm{kl}}(p^\beta\|q^\beta) + D_{\mathrm{kl}}(q^\beta\|p^\beta))^{1/2} + O(\beta)$$
$$\leq (W_\mathcal{F}(p^\beta, q^\beta) + O(\beta \log(1/\beta)))^{1/2} + O(\beta) \leq O(\tilde{d}_\mathcal{F}(p, q)) \tag{34}$$

Thus we prove the lower bound.

Proceeding to the upper bound, notice that $W_\mathcal{F}(p^\beta, q^\beta) \lesssim \frac{1}{\beta^4} W_1(p^\beta, q^\beta)$ since every function in $\mathcal{F}$ is $O(\mathrm{poly}(d)\frac{1}{\beta^4})$-Lipschitz by Theorem E.2. We relate $W_1(p^\beta, q^\beta)$ to $W_1(p, q)$, more precisely we prove: $W_1(p^\beta, q^\beta) \leq (1 + e^{-d})W_1(p, q)$. Having this, we'd be done: namely, we simply set $\beta = W^{1/6}$ to get the necessary bound.

Proceeding to the claim, consider the optimal coupling $C$ of $p, q$, and consider the induced coupling $C_z$ on the latent variable $z$ in $p, q$. Then,

$$W_1(p, q) = \int_{z \in \mathbb{R}^d} \|G(z) - G(z')\|_1 dC_z(z, z') \det\left(\frac{\partial G_\theta(z)}{\partial z}\right) \det\left(\frac{\partial G_\theta(z')}{\partial z'}\right)$$

Consider the coupling $\tilde{C}_z$ on the latent variables of $p^\beta, q^\beta$, specified as $\tilde{C}_z(z, z') = C(z, z')(1 - \Pr[z \notin D_z])^2$. The coupling $\tilde{C}$ of $p^\beta, q^\beta$ specified by coupling $z$'s according to $\tilde{C}_z$ and the (truncated) Gaussian noise to be the same in $p^\beta, q^\beta$, we have that

$$W_1(p^\beta, q^\beta) \leq \int_{z \in D_z} \|G(z) - G(z')\|_1 d\tilde{C}_z(z, z') \det\left(\frac{\partial G_\theta(z)}{\partial z}\right) \det\left(\frac{\partial G_\theta(z')}{\partial z'}\right)$$
$$\leq \int_{z \in D_z} (1 + e^{-d})\|G(z) - G(z')\|_1 dC_z(z, z') \det(\frac{\partial G_\theta(z)}{\partial z}) \det\left(\frac{\partial G_\theta(z')}{\partial'}\right)$$
$$\leq (1 + e^{-d})W_1(p, q)$$

The generalization claim follows completely analogously to Lemma D.5, using the Lipschitzness bound of the generators in Theorem E.2.

$\square$

The rest of the section is dedicated to the proof of Theorem E.2, which will be finally in Section E.3.

### E.2 Tools and helper lemmas

First, we prove several helper lemmas:

**Lemma E.3** (Quantitative bijectivity). $\|G(\tilde{z}) - G(z)\| \geq \frac{1}{L_\sigma^l} \prod_{j=i}^{L} \sigma_{\min}(W_j)\|\tilde{z} - z\|.$

*Proof.* The proof proceeds by reverse induction on $l$. We will prove that

$$\|h_i - \tilde{h}_i\| \geq \frac{1}{L_\sigma^{l-i}} \prod_{j=i}^{L} \sigma_{\min}(W_j)\|\tilde{z} - z\|$$

The claim trivial holds for $i = 0$, so we proceed to the induction. Suppose the claim holds for $i$. Then,

$$\|W_i h_i + b_i - (W_i \tilde{h}_i + b_i)\| \geq \frac{1}{\sigma_{\min}(W_i)} \|h_i - \tilde{h}_i\|$$

and

$$\|\sigma(W_i h_i + b_i) - \sigma(W_i \tilde{h}_i + b_i)\| \geq \frac{L_\sigma}{\sigma_{\min}(W_i)} \|h_i - \tilde{h}_i\|$$

by Lipschitzness of $\sigma^{-1}$. Since $h_{i-1} = \sigma(W_i h_i + b_i)$ and $\tilde{h}_{i-1} = \sigma(W_i h_i + b_i)$,

$$\|h_{i-1} - \tilde{h}_{i-1}\| \geq \frac{1}{L_\sigma^{l-(i-1)}} \prod_{j=i-1}^{L} \sigma_{\min}(W_j) \|\tilde{z} - z\|$$

as we need. □

**Lemma E.4** (Approximate inversion). *Let $x \in \mathbb{R}^d$ be s.t. $\exists z, \|G_\theta(z) - x\| \leq \epsilon$. Then, there is a neural network $N$ of size $O(ld^2)$, activation function $\sigma^{-1}$ and Lipschitz constant $L_\sigma^l \prod_{i=1}^{l} \frac{\sigma_{\max}(W_i)}{\sigma_{\min}^2(W_i)}$ which recovers a $\hat{z}$, s.t. $\|\hat{z} - z\| \leq \epsilon 2^l L_\sigma^l \prod_{i=1}^{l} \frac{1}{\sigma_{\min}(W_i)}$*

*Proof.* $N$ will iteratively produce estimates $\hat{h}_i$, s.t.
(1) $\hat{h}_0 = x$
(2) $\hat{h}_i = \sigma^{-1}(\text{argmin}_h \|W_i h + b_i - \sigma^{-1}(\hat{h}_{i-1})\|_2^2)$

We will prove by induction that $|h_i - \hat{h}_i| \leq \epsilon 2^i L_\sigma^i \prod_{j=1}^{i} \frac{1}{\sigma_{\min}(W_j)}$. The claim trivial holds for $i = 0$, so we proceed to the induction. Suppose the claim holds for $i$. Then,

$$\begin{aligned}
\min_h \|W_{i+1} h + b_{i+1} - \hat{h}_i\| &\leq \|W_{i+1} h_{i+1} + b_{i+1} - \sigma^{-1}(\hat{h}_i)\| \\
&= \|\sigma^{-1}(h_i) - \sigma^{-1}(\hat{h}_i)\| \\
&\leq L_\sigma \|h_i - \hat{h}_i\| \\
&\leq \epsilon 2^i L_\sigma^{i+1} \prod_{j=1}^{i} \frac{1}{\sigma_{\min}(W_j)}
\end{aligned}$$

where the last inequality holds by the inductive hypothesis, and the next-to-last one due to Lipschitzness of $\sigma^{-1}$.

Hence, denoting $\tilde{h} = \text{argmin}_h \|W_{i+1} h + b_{i+1} - \sigma^{-1}(\hat{h}_i)\|_2^2$, we have

$$\begin{aligned}
\|W_{i+1}\tilde{h} - W{i+1}h_{i+1})\| &= \|W_{i+1}\tilde{h} + b_{i+1} - \sigma^{-1}(\hat{h}_i) + \sigma^{-1}(\hat{h}_i) - W_{i+1}h_{i+1} - b_{i+1}\| \\
&\leq \|W_{i+1}\tilde{h} + b_{i+1} - \sigma^{-1}(\hat{h}_i)\| + \|\sigma^{-1}(\hat{h}_i) - W_{i+1}h_{i+1} - b_{i+1}\| \\
&= \|W_{i+1}\tilde{h} + b_{i+1} - \sigma^{-1}(\hat{h}_i)\| + \|\sigma^{-1}(\hat{h}_i) - \sigma^{-1}(h_i)\| \\
&\leq \epsilon 2^{i+1} L_\sigma^{i+1} \prod_{j=1}^{i} \frac{1}{\sigma_{\min}(W_j)}
\end{aligned}$$

This implies that

$$\|W_{i+1}(\tilde{h} - h_{i+1})\| \leq 2\epsilon L_\sigma^{i+1} \prod_{j=1}^{i} \frac{1}{\sigma_{\min}(W_j)}$$

which in turns means

$$\|\tilde{h} - h_{i+1}\| \leq \epsilon 2^{i+1} L_\sigma^{i+1} \prod_{j=1}^{i+1} \frac{1}{\sigma_{\min}(W_j)}$$

which completes the claim.

Turning to the size/Lipschitz constant of the neural network: all we need to notice is that $\hat{h}_i = \sigma^{-1}(W_i^\dagger(\hat{h}_{i-1} - b_i))$, which immediately implies the Lipschitzness/size bound by simple induction.

$\square$

We also introduce a few tools to get a handle on functions that can be approximated efficiently by neural networks of small size/Lipschitz constant.

**Lemma E.5** (Composing Lipschitz functions). *If $f : \mathbb{R}^{d_2} \to \mathbb{R}^{d_3}$ and $g : \mathbb{R}^{d_1} \to \mathbb{R}^{d_2}$ are $L, K$-Lipschitz functions respectively, then $f \circ g : \mathbb{R}^{d_1} \to \mathbb{R}^{d_3}$ is $LK$-Lipschitz.*

*Proof.* The proof follows by definition essentially:
$$\|f(g(x)) - f(g(x'))\| \leq L\|g(x) - g(x')\| \leq LK\|x - x'\|$$

$\square$

**Lemma E.6** (Calculating singular value decomposition approximately, (Demmel et al., 2007)). *There is a neural network with size $O(n^3 poly(\log(1/\epsilon)))$ that given a symmetric matrix $A \in \mathbb{R}^{n \times n}$ with minimum eigenvalue gap $\min_{i \neq j} |\lambda_i - \lambda_j| \geq \delta$ and eigenvectors $\{u_i\}$ outputs $\{\tilde{u}_i, \tilde{\lambda}_i\}$ s.t. :*
*(1) $|\langle \tilde{u}_i, \tilde{u}_j \rangle| \leq \epsilon, \forall i \neq j$ and $\|\tilde{u}_i\| = 1 \pm \epsilon$.*
*(2) $|\tilde{u}_i - u_i| \leq \epsilon/\delta, |\tilde{\lambda}_i - \lambda_i| \leq \epsilon, \forall i$.*
*(Note the eigenvalue/eigenvector pairs for $A$ are unique since the minimum eigenvalue gap is nonzero).*

**Lemma E.7** (Backpropagation, (Rumelhart et al., 1986)). *Given a neural network $f : \mathbb{R}^m \to \mathbb{R}$ of depth $l$ and size $N$, there is a neural network of size $O(N + l)$ which calculates the gradient $\frac{\partial f}{\partial i}, i \in [m]$.*

### E.3 PROOF OF THEOREM E.2

We will proceed to prove the two parts one at a time.

First, we prove the following lemma, which can be seen as a quantitative version of Laplace's method for evaluating integrals:

**Lemma E.8** ("Tail" bound for integral at $z^*$). *Let $x = G(z^*) + r$, for $\|r\| \leq 10\beta\sqrt{d}\log d$, and $\|z^*\| \leq 10\sigma\sqrt{d}\log d$. The, for $\beta = O(poly(1/d))$, and*
$$\delta = 100\beta \log(1/\beta)\frac{\sqrt{d}}{R}$$

*it holds that*
$$\int_{z:\|z-z^*\|>\delta, z \in D_z} e^{f(z)} dz \leq \beta \int_{z \in D_z} e^{f(z)} dz$$

*Proof.* Let's write
$$\int_{z \in D_z} e^{f(z)} dz = \int_{z:\|z-z^*\|\leq\delta} e^{f(z)} dz + \int_{z:\|z-z^*\|>\delta, z \in D_z} e^{f(z)} dz \tag{35}$$
To prove the claim of the Lemma, we will lower bound the first term, and upper bound the latter, from which the conclusion will follow.

Consider the former term. Taylor expanding in a neighborhood around $z^*$ we have
$$f(z) = f(z^*) + (z - z^*)^\top \nabla^2 f(z^*)(z - z^*) \pm \frac{T}{\beta^2}\|z - z^*\|^3$$

where the first-order term vanishes since $z^*$ is a global optimum. Furthermore, $\nabla^2 f(z^*) \preceq 0$ for the same reason. Hence, by Taylor's theorem with remainder bounds, and using the fact that $e^x \geq 1 + x$, we have
$$\int_{z:\|z-z^*\|\leq\delta} e^{f(z)} dz \geq \left(1 - \frac{T}{\beta^2}\delta^3\right) f(z^*) \int_{z:\|z-z^*\|\leq\delta} (z - z^*)^\top \nabla^2 f(z^*)(z - z^*)$$

The integral on the right is nothing more than the (unnormalized) cdf of a Gaussian with covariance matrix $(-\nabla^2 f(z^*))^{-1}$.

Moreover, $-\nabla^2 f(z^*)$ is positive definite with smallest eigenvalue bounded by $\frac{R^2}{d\beta^2}$, since

$$-\nabla^2 f(z^*) = \nabla^2(\|z^*\|^2) + \frac{1}{\beta^2}\nabla^2(\|G(z^*) - x\|^2)$$

$$= I + \frac{1}{\beta^2}\nabla^2(\|G(z^*) - x\|^2)$$

and

$$\nabla^2(\|G(z^*) - x\|^2) \succeq \sum_i \nabla G_i(z^*)\nabla^\top G_i(z^*) + \sum_i (G_i(z) - x_i)\nabla^2 G_i(z)$$

where $G_i$ is the $i$-th coordinate of $G$. We claim $\sum_i \nabla G_i(z^*)\nabla^\top G_i(z^*) \succeq \frac{R^2}{d}I$, and $\|\sum_i (G_i(z) - x_i)\nabla^2 G_i(z)\|_2 \lesssim \beta\sqrt{d}\log d$. The latter follows from the bound on $r$ and Cauchy-Schwartz. For the former, note that we have

$$v^\top \left(\sum_i \nabla G_i(z^*)\nabla^\top G_i(z^*)\right) v = \sum_i \langle v, \nabla G_i(z^*)\rangle^2$$

$$= \sum_i \left(\lim_{\epsilon\to 0} \frac{G_i(z^* + \epsilon v) - G_i(z^*)}{\epsilon}\right)^2$$

By Lemma E.3, $\|G(z^* + \epsilon v) - G(z^*)\| \geq R\epsilon$, so $\exists i$, s.t. $|G_i(z^* + \epsilon v) - G_i(z^*)| \geq \frac{R\epsilon}{\sqrt{d}}$. Hence,

$$\sum_i \left(\lim_{\epsilon\to 0} \frac{G_i(z^* + \epsilon v) - G_i(z^*)}{\epsilon}\right)^2 \geq \frac{R^2}{d}$$

from which $\sum_i \nabla G_i(z^*)\nabla^\top G_i(z^*) \succeq \frac{R^2}{d}$ follows.

Using standard Gaussian tail bounds, since $\delta \geq \frac{\beta\log(1/\beta)}{R}\sqrt{d}$, we have

$$\int_{z:\|z-z^*\|\leq\delta} (z - z^*)^\top \nabla^2 f(z^*)(z - z^*) \geq (1 - \beta)\det(4\pi(-\nabla^2 f(z^*)))^{1/2} \qquad (36)$$

We proceed to the latter term in eq. (35). We have

$$f(z^*) - f(z) = \left|\frac{\|z^*\|^2 + \|G(z^*) - x\|^2}{\beta^2} - \|z\|^2 - \frac{\|G(z) - x\|^2}{\beta^2}\right|$$

$$\geq \left|\frac{\|G(z^*) - x\|^2}{\beta^2} - \frac{\|G(z) - x\|^2}{\beta^2}\right| - \left|\|z\|^2 - \|z^*\|^2\right|$$

$$= \left|\frac{\|G(z^*) - G(z^*) - r\|^2}{\beta^2} - \frac{\|G(z) - G(z^*) + r\|^2}{\beta^2}\right| - \left|\|z\|^2 - \|z^*\|^2\right|$$

$$\overset{\text{①}}{\geq} \frac{(R\|z - z^*\| - \|r\|)^2}{\beta^2} - \frac{\|r\|^2}{\beta^2} - \left|\|z\|^2 - \|z^*\|^2\right|$$

$$\overset{\text{②}}{\geq} \frac{(R\|z - z^*\| - \|r\|)^2}{\beta^2} - \frac{\|r\|^2}{\beta^2} - \|z - z^*\|^2 - 2\|z - z^*\|\|z^*\|$$

where ① follows from the definition of $x$, and ② by triangle inequality.

Note also that $\frac{1}{2}R\|z - z^*\| \geq \|r\|$ since $\delta \geq \frac{2\|r\|}{R}$, which in turn implies

$$\frac{(R\|z - z^*\| - \|r\|)^2}{\beta^2} - \frac{\|r\|^2}{\beta^2} - \|z - z^*\|^2 - 2\|z - z^*\|\|z^*\| \geq \frac{3}{16}\frac{R^2}{\beta^2}\|z - z^*\|$$

Finally, $\beta^2 \leq \frac{R^2}{32}\|z^*\|$, so that it follows

$$\|z - z^*\|^2 \left(\frac{3R^2}{16\beta^2} - 1\right) - 2\|z - z^*\|\|z^*\| \leq \frac{3R^2}{32\beta^2}\|z - z^*\|^2$$

Putting these estimates together, we get $f(z) < f(z^*) - \frac{3R^2}{32\beta^2}\|z - z^*\|^2$, which implies

$$\int_{z:\|z-z^*\|>\delta} e^{f(z)} dz \leq e^{f(z^*)} \int_{z:\|z-z^*\|>\delta} e^{-\frac{3R^2}{32\beta^2}\|z-z^*\|^2}$$

The integral on the right is again the unnormalized cdf of a Gaussian with covariance matrix $\frac{3R^2}{32\beta^2} I$, so by Gaussian tail bounds again, and using that the smallest eigenvalue of $-\nabla^2 f(z^*)$ is lower-bounded by $\frac{R^2}{\beta^2}$ we have

$$\int_{z:\|z-z^*\|>\delta} e^{f(z)} \leq \beta e^{f(z^*)} \det(4\pi(-\nabla^2 f(z^*)))^{1/2}$$

as we wanted.

Putting this together with eq. (36), we get the statement of the theorem.

$\square$

With this in mind, we can prove part (1) of our Theorem E.2 restated below:

**Theorem E.9.** *There is a neural network $N$ of size poly$(\frac{1}{\beta}, R, L_G, T, S, d)$ with Lipschitz constant $O(\text{poly}(R, L_G, d, T, S), 1/\beta^4)$ which given as input $x = G(z^*) + r$, for $\|r\| \leq 10\beta\sqrt{d}\log d$, and $\|z^*\| \leq 10\sqrt{d}\log d$ for $\beta = O(\text{poly}(1/d))$ outputs $N(x)$, s.t.*

$$|N(x) - \log p^\beta(x)| = O_{poly(d)}(\beta \log(1/\beta)) + \exp(-d)$$

*Proof.* It suffices to approximate

$$\int_{z \in D_z} e^{f(z)} dz \tag{37}$$

up to a multiplicative factor of $1 \pm O_{\text{poly}(d)}(\beta \log(1/\beta))$, since the normalizing factor satisfies

$$\int_{x \in D_x} \int_{z \in D_z} e^{-\|z\|^2} e^{-\frac{\|G_\theta(z)-x\|^2}{\beta^2}} dz = (1 \pm \exp(-d))\det(4\pi I)^{-1/2}\det(4\pi/\beta^2 I)^{-1/2}$$

We will first present the algorithm, then prove that it:
(1) Approximates the integral as needed.
(2) Can be implemented by a small, Lipschitz network as needed.

The algorithm is as follows:

---

**Algorithm 1** Discriminator family with restricted approximability for degenerate manifold

---

1: **Parameters:** Matrices $E_1, E_2, \ldots, E_r \in \mathbb{R}$, matrices $W_1, W_2, \ldots, W_l$.
2: Let $\delta = \frac{100\frac{L_G}{\beta}\log(1/\beta)}{R^2}$ and let $\mathcal{S}$ be the trivial $\beta^2$-net of the matrices with spectral norm bounded by $O(1/\beta^2)$.
3: Let $\hat{z} = N_{\text{inv}}(x)$ be the output of the "invertor" circuit of Lemma E.4.
4: Calculate $g = \nabla f(\hat{z}), H = \nabla^2 f(\hat{z})$ by the circuit implied in Lemma E.7.
5: Let $M$ be the nearest matrix in $\mathcal{S}$ to $H$ and $E_i, i \in [r]$ be s.t. $M + E_i$ has $\Omega(\beta)$-separated eigenvalues. (If there are multiple $E_i$ that satisfy the separation condition, pick the smallest $i$.)
6: Let $(e_i, \lambda_i)$ be approximate eigenvector/eigenvalue pairs of $H + E_i$ calculated by the circuit implied in Lemma E.6.
7: Approximate $I_i = \log \left( \int_{|c_i| \leq \delta} e^{c_i \langle e_i, g \rangle + \sum_i c_i^2 \lambda_i} dc_i \right), i \in [r]$ by subdividing the interval $(0, \delta)$ into intervals of size $\beta^2$ and evaluating the resulting Riemannian sum instead of the integral.
8: Output $\sum_i I_i$.

---

First, we will show (1), namely that the Algorithm 1 approximates the integral of interest. We'll use an approximate version of Lemma E.8 – with a slightly different division of where the "bulk" of the integral is located. As in Algorithm 1, let $\hat{z} = N_{\mathbf{inv}}(x)$ be the output of the "invertor" circuit of Lemma E.4. and let $\delta = \frac{100d\frac{L_G}{\beta}\log(1/\beta)}{R^2}$ and denote by $B$ the set $B = \{z : |\langle z - \hat{z}, e_i\rangle| \leq \delta\}$. Furthermore, let's define how the matrices $E_i, i \in [r]$ are to be chosen. Let $\mathcal{S}$ be an $\beta^2$-net of the matrices with spectral norm bounded by $O(1/\beta^2)$. We claim that there exist matrices $E_1, E_2, \ldots, E_r, r = \Omega(d\log(1/\beta))$, s.t. if $M \in \mathcal{S}$, at least one of the matrices $M + E_i, i \in [r]$ has eigenvalues that are $\Omega(\beta)$-separated and $\|E_i\|_2 \leq \frac{\sqrt{d}}{\beta}$. Indeed, let let $E$ be a random Gaussian matrix with entrywise variance $\frac{1}{\beta^2}$. By Theorem 2.6 in (Nguyen et al., 2017), for any fixed matrix $A$, with probability $3/4$, $\min_i |\lambda_i(A + E) - \lambda_{i+1}(A + E)| = \Omega(\beta)$. The number of matrices in $\mathcal{S}$ is bounded by $2^{O(d\log(1/\beta))}$, so $E_i, i \in [r]$ exist by the probabilistic method.

We can write the integral of interest as

$$\int_{z \in B} e^{f(z)}dz + \int_{z \in \bar{B}} e^{f(z)}dz$$

Note that

$$\begin{aligned}\|z - z^*\| &= \|z - \hat{z} + \hat{z} - z^*\| \\ &\geq \|z - \hat{z}\| - \|\hat{z} - z^*\| \\ &\geq \frac{4L_G}{R}\|\hat{z} - z\|\end{aligned}$$

This means that $\{z : \|z - z^*\| \leq \frac{4L_G}{R}\delta\} \subseteq B$, so by Lemma E.8, we have

$$\int_{z \in \bar{B}} e^{f(z)} \leq O(\beta)\int_z e^{f(z)}dz$$

which means that to prove the statement of the Theorem, it suffices for us to approximate $\int_{z \in B} e^{f(z)}dz$.

Consider the former term first. By Taylor's theorem with remainder, expanding $f$ in a $\delta$-neighborhood near $\hat{z}$, we get

$$f(z) = f(\hat{z}) + (z - \hat{z})^\top \nabla f(\hat{z}) + (z - \hat{z})^\top \nabla^2 f(\hat{z})(z - \hat{z}) \pm \frac{T}{\beta^2}\|z - \hat{z}\|^3$$

For notational convenience, same as in Algorithm 1, let's denote by $H := (z - \hat{z})^\top \nabla^2 f(\hat{z})(z - \hat{z})$, and $g := \nabla f(\hat{z})$. Steps 5-8 effectively perform a change of basis in the eigenbasis of $H$ and evaluate the integral in this basis – however to ensure Lipschitzness (which we prove later on), we will need to perturb $H$ slightly.

Let $M$ be the closest matrix to $\nabla^2 f(\hat{z})$ in the $\beta^2$-net $\mathcal{S}$ and let $e_i$ be approximate eigenvectors of $\tilde{H} = M + E_i$ in the sense of Lemma E.6, s.t. the eigenvalues of $M + E_i$ are $\Omega(\beta)$-separated.

Since $\|E_i\| \leq \frac{\sqrt{d}}{\beta}$, we have

$$|(z - \hat{z})^\top \tilde{H}(z - \hat{z}) - (z - \hat{z})^\top H(z - \hat{z})| = O(\beta)$$

Hence,

$$f(z) = f(\hat{z}) + (z - \hat{z})^\top \nabla f(\hat{z}) + (z - \hat{z})^\top \tilde{H}(z - \hat{z}) \pm \frac{T}{\beta^2}\|z - \hat{z}\|^3 \pm O(\beta)$$

Towards rewriting $f$ in the approximate basis $e_i$, let $z - \hat{z} = \sum_i c_i e_i$ for some scalars $c_i$. We have

$$f(z) = f(\hat{z}) + \sum_i c_i\langle e_i, g\rangle + \sum_i c_i^2 \lambda_i \pm \frac{T}{\beta^2}(\sum_i c_i^2)^{3/2}$$

By Taylor's theorem with remainder, $e^x = 1 + x \pm e\frac{x^2}{2}$, if $x < 1$. Hence,

$$\int_{z \in B} e^{f(z)} dz = \left(1 \pm ed^{3/2} \frac{T}{\beta^2} \delta^3\right) e^{f(\hat{z})} \prod_i \int_{|c_i| \leq \delta} e^{c_i \langle e_i, g \rangle + \sum_i c_i^2 \lambda_i}$$

Calculating the integral by subdividing $(0, \delta)$ into intervals of size $\beta^2$, and approximating the integral by the accompanying Riemannian sum, and taking into account $|c_i| = O(\beta)$ and $\lambda_i, \|g\| = O(\frac{1}{\beta^2})$, we get a multiplicative approximation of

$$\int_{|c_i| \leq \delta} e^{c_i \langle e_i, g \rangle + \sum_i c_i^2 \lambda_i}$$

of the order $e^\beta = 1 + O(\beta)$, which is what we want.

We turn to implementing the algorithm by a small neural network with good Lipschitz constant. Both the Lipschitz constant and the size will be handled by the composition Lemma E.5 and analyzing each step of Algorithm 1. Steps 3 and 4 are handled by our helper lemmas: the invertor circuit by Lemma E.4 has Lipschitz constant $L_G$; calculating the Hessian $\nabla^2 f(\hat{z})$ can be performed by a polynomially sized neural network by Lemma E.7 and since the third partial derivatives are bounded, so the output of this network is $O(\frac{T}{\beta^2})$-Lipschitz as well. We turn to the remaining steps.

Proceeding to the eigendecomposition, by Lemma E.6, we can perform an approximate eigendecomposition of $\tilde{H}$ with a network of size $O(\text{poly}(d))$ – so we only need to handle the Lipschitzness. We will show that the result of Steps 5-6, the vectors $u_j$ and scalars $\lambda_j$ are Lipschitz functions of $H$.

Suppose that $H$ and $H'$ are s.t. $\|H - H'\| \leq \beta^2$ first. Then, $H, H'$ are mapped to the same matrix $M$ in $\mathcal{S}$, $\{u_j\}$ and $\{u_j'\}$ are the eigenvectors of $H + E_i$ and $H' + E_i$ for some $i \in [r]$. First, by Weyl's theorem, since $H + E_i = M + (H - M) + E_i$, the eigenvalues of $H + E_i$ are $\Omega(\beta) - \|H - M\| = \Omega(\beta)$-separated. Furthermore, since $H' + E_i = H + E_i + (H' - H)$, by Wedin's theorem, $\|u_j(H' + E_i) - u_j(H + E_i)\| = O\left(\|H - H'\|_2 \frac{1}{\beta}\right)$. If, on the other hand, $\|H - H'\| > \beta^2$, and $E_a, E_b$ are the perturbation matrices used for $H$ and $H'$, since the eigenvalues of $H + E_a$ are $\Omega(\beta)$-separated, by Wedin's theorem,

$$\|u_j(H' + E_b) - u_j(H + E_a)\| \leq O\left(\frac{1}{\beta} \left(\|H - H'\|_2 + \|E_b - E_a\|_2\right)\right) = O\left(\frac{1}{\beta} \|H - H'\|_2\right)$$

so we get that the map to the vectors $u_j$ is $O(1/\beta)$-Lipschitz. A similar analysis shows that the map to the eigenvalues $\lambda_j$ is also Lipschitz.

Finally, we move to calculating the integral in Step 7: the trivial implementation of the integral by a neural network has size $O(\text{poly}(\beta))$ and as a function of $e_i, g, \lambda_i$ is $O(1/\beta)$-Lipschitz, which proves the statement of the theorem.

$\square$

Moving to Part (2) of Theorem E.2, we prove:

**Theorem E.10.** *Let $N$ be the neural network $N$ used in Theorem E.9. The network additionally satisfies $N(x) \leq \log p(x) + O_{R,L_G,d}(\beta \log(1/\beta)) + \exp(-d)$, $\forall x \in D$.*

*Proof.* Recalling the proof of Theorem E.9, and reusing the notation there, we can express

$$q(x) = \int_z \exp(f(z)) = \int_{z \in B} e^{f(z)} dz + \int_{z \in \bar{B}} e^{f(z)} dz$$

Since the neural network $N$ ignores the latter term, and we need only produce an upper bound, it suffices to show that $N$ approximates

$$\int_{z: \|z - \hat{z}\| \leq \delta} e^{f(z)} dz$$

up to a multiplicative factor of $1 - O(\beta \log(1/\beta))$. However, if we consider the proof of Theorem E.9, we notice that the approximation consider there indeed serves our purpose: Taylor-expanding same as there, we have

$$\int_{z \in B} e^{f(z)} dz = \left(1 \pm e d^{3/2} T_\beta \delta^3\right) e^{f(\hat{z})} \prod_i \int_{|c_i| \leq \delta} e^{c_i \langle e_i, g \rangle + \sum_i c_i^2 \lambda_i}$$

This integral can be evaluated in the same manner as in Theorem E.9, as our bound on $T_\beta$ holds universally on neighborhood of radius $D_x$.

$\square$

Finally, part (3) follows easily from (1) and (2):

*Proof of Part 3 of Theorem E.2.* For points $x$, s.t. $\nexists z, \|G(z) - x\| \leq 10\beta\sqrt{d} \log d$, it holds that $p(x) = O(\exp(-d))$. On the other hand, by the Lipschitzness of $N$, we have $\|N(x)\| = O_{\frac{1}{\beta^4}}(\|x\|)$. Since $x \in D_x$ implies $\|x\| = O(\text{poly}(d))$ the claim follows. $\square$

## F  EXPERIMENTS ON SYNTHETIC 2D DATASETS: UNIT CIRCLE

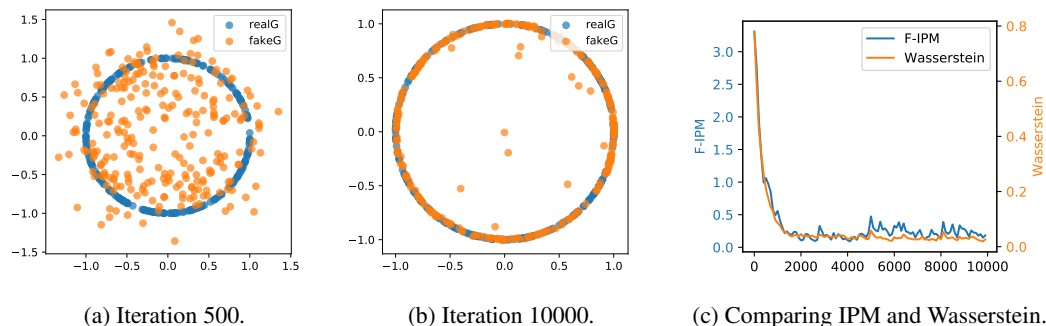

(a) Iteration 500.     (b) Iteration 10000.     (c) Comparing IPM and Wasserstein.

Figure 2: Experiments on the unit circle dataset. The neural net IPM, the Wasserstein distance, and the sample quality are correlated along training. (a)(b): Sample batches from the ground truth and the learned generator at iteration 500 and 10000. (c): Comparing the F-IPM and the Wasserstein distance. RealG and fakeG denote the ground truth generator and the learned generator, respectively.

## G  EXPERIMENTS ON INVERTIBLE NEURAL NET GENERATORS

We further perform synthetic WGAN experiments with invertible neural net generators (cf. Section 4.1) and discriminators designed with restricted approximability (Lemma 4.1). In this case, the invertibility guarantees that the KL divergence can be computed, and our goal is to demonstrate that the empirical IPM $W_{\mathcal{F}}(p, q)$ is well correlated with the KL-divergence between $p$ and $q$ on synthetic data for various pairs of $p$ and $q$ (The true distribution $p$ is generated randomly from a ground-truth neural net, and the distribution $q$ is learned using various algorithms or perturbed version of $p$.)

### G.1  SETUP

**Data**  The data is generated from a ground-truth invertible neural net generator (cf. Section 4.1), i.e. $X = G_\theta(Z)$, where $G_\theta : \mathbb{R}^d \to \mathbb{R}^d$ is a $\ell$-layer layer-wise invertible feedforward net, and $Z$ is a spherical Gaussian. We use the Leaky ReLU with negative slope $0.5$ as the activation function $\sigma$, whose derivative and inverse can be very efficiently computed. The weight matrices of the layers are set to be well-conditioned with singular values in between $0.5$ to $2$.

We choose the discriminator architecture according to the design with restricted approximability guarantee (Lemma 4.1, eq. (10) eq. (11)). As $\log \sigma^{-1'}$ is a piecewise constant function that is not

differentiable, we instead model it as a trainable one-hidden-layer neural network that maps reals to reals. We add constraints on all the parameters in accordance with Assumption 1.

**Training** To train the generator and discriminator networks, we generate stochastic batches (with batch size 64) from both the ground-truth generator and the trained generator, and solve the min-max problem in the Wasserstein GAN formulation. We perform 10 updates of the discriminator in between each generator step, with various regularization methods for discriminator training (specified later). We use the RMSProp optimizer (Tieleman & Hinton, 2012) as our update rule.

**Evaluation metric** We evaluate the following metrics between the true and learned generator.

(1) The KL divergence. As the density of our invertible neural net generator can be analytically computed, we can compute their KL divergence from empirical averages of the difference of the log densities:

$$\widehat{D_{\mathrm{kl}}}(p^\star, p) = \mathbb{E}_{X \sim \widehat{p^\star}^n}[\log p^\star(X) - \log p(X)],$$

where $p^\star$ and $p$ are the densities of the true generator and the learned generator. We regard the KL divergence as the "correct" and rather strong criterion for distributional closeness.

(2) The training loss (IPM $W_{\mathcal{F}}$ train). This is the (unregularized) GAN loss during training. Note: as typically in the training of GANs, we balance carefully the number of steps for discriminator and generators, the training IPM is potentially very far away from the true $W_{\mathcal{F}}$ (which requires sufficient training of the discriminators).

(3) The neural net IPM ($W_{\mathcal{F}}$ eval). We report once in a while a separately optimized WGAN loss in which the learned generator is held fixed and the discriminator is trained from scratch to optimality. Unlike the training loss, here the discriminator is trained in norm balls but with no other regularization. By doing this, we are finding $f \in \mathcal{F}$ that maximizes the contrast and we regard the $f$ found by stochastic optimization an approximate maximizer, and the loss obtained an approximation of $W_{\mathcal{F}}$.

Our theory shows that for our choice of $\mathcal{G}$ and $\mathcal{F}$, WGAN is able to learn the true generator in KL divergence, and the $\mathcal{F}$-IPM (in evaluation instead of training) should be indicative of the KL divergence. We test this hypothesis in the following experiments.

### G.2 CONVERGENCE OF GENERATORS IN KL DIVERGENCE

In our first experiment, $G$ is a two-layer net in $d = 10$ dimensions. Though the generator is only a shallow neural net, the presence of the nonlinearity makes the estimation problem non-trivial. We train a discriminator with the architecture specified in Lemma 4.1), using either Vanilla WGAN (clamping the weight into norm balls) or WGAN-GP (Gulrajani et al., 2017) (adding a gradient penalty). We fix the same ground-truth generator and run each method from 6 different random initializations. Results are plotted in Figure 3.

Our main findings are two-fold:

(1) WGAN training with discriminator design of restricted approximability is able to learn the true distribution in KL divergence. Indeed, the KL divergence starts at around 10 - 30 and the best run gets to KL lower than 1. As KL is a rather strong metric between distributions, this is strong evidence that GANs are finding the true distribution and mode collapse is not happening.

(2) The $W_{\mathcal{F}}$ (eval) and the KL divergence are highly correlated with each other, both along each training run and across different runs. In particular, adding gradient penalty improves the optimization significantly (which we see in the KL curve), and this improvement is also reflected by the $W_{\mathcal{F}}$ curve. Therefore the quantity $W_{\mathcal{F}}$ can serve as a good metric for monitoring convergence and is at least much better than the training loss curve.

To test the necessity of the specific form of the discriminator we designed, we re-do the same experiment with vanilla fully-connected discriminator nets. Results (in Appendix G.4) show that IPM with vanilla discriminators also correlate well with the KL-divergence. This is not surprising from

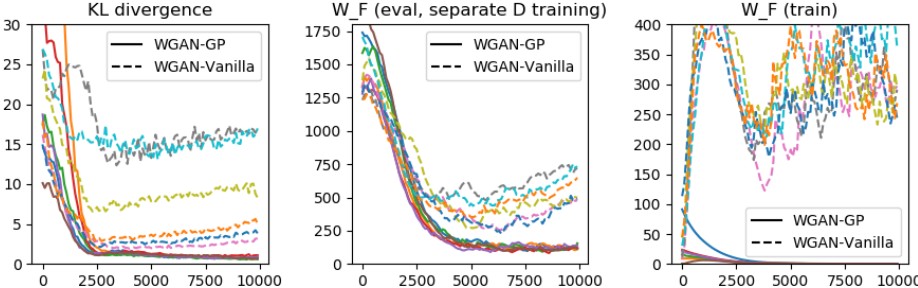

Figure 3: Learning an invertible neural net generator on synthetic data. The x-axis in all the graphs indicates the number of steps. The left-most figure shows the KL-divergence between the true distribution $p$ and learned distribution $q$ at different steps of training, the middle the estimated IPM (evaluation) between $p$ and $q$, and the right one the training loss. We see that the estimated IPM in evaluation correlates well with the KL-divergence. Moving average is applied to all curves.

a theoretical point of view because a standard fully-connected discriminator net (with some over-parameterization) is likely to be able to approximate the log density of the generator distributions (which is essentially the only requirement of Lemma 4.3.)

For this synthetic case, we can see that the inferior performance in KL of the WGAN-Vanilla algorithm doesn't come from the statistical properties of GANs, but rather the inferior training performance in terms of the convergence of the IPM. We conjecture similar phenomenon occurs in training GANs with real-life data as well.

## G.3    PERTURBED GENERATORS

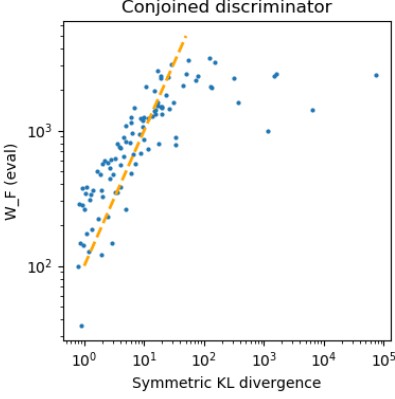

Figure 4: Scatter plot of KL divergence and neural net IPM on perturbed generator pairs. Correlation between $\log(D_{\mathrm{kl}}(p\|q) + D_{\mathrm{kl}}(q\|p))$ and $\log W_{\mathcal{F}}$ is 0.7315. Dashed line is $W_{\mathcal{F}}(p,q) = 100(D_{\mathrm{kl}}(p\|q) + D_{\mathrm{kl}}(q\|p))$.

In this section, we remove the effect of the optimization and directly test the correlation between $p$ and its perturbations. We compare the KL divergence and neural net IPM on pairs of *perturbed generators*. In each instance, we generate a pair of generators $(G, G')$ (with the same architecture as above), where $G'$ is a perturbation of $G$ by adding small Gaussian noise. We compute the KL divergence and the neural net IPM between $G$ and $G'$. To denoise the unstable training process for computing the neural net IPM, we optimize the discriminator from 5 random initializations and pick the largest value as the output.

As is shown in Figure 4, there is a clear positive correlation between the (symmetric) KL divergence and the neural net IPM. In particular, majority of the points fall around the line $W_{\mathcal{F}} = 100 D_{\mathrm{kl}}$, which is consistent with our theory that the neural net distance scales linearly in the KL divergence.

Note that there are a few outliers with large KL. This happens mostly due to the perturbation being accidentally too large so that the weight matrices become poorly conditioned – in the context of our theory, they fall out of the good constraint set as defined in Assumption 1.

### G.4 EXPERIMENTS WITH VANILLA DISCRIMINATOR

#### G.4.1 CONVERGENCE OF GENERATORS IN KL DIVERGENCE

We re-do the experiments of Section G.2 with vanilla fully-connected discriminator nets. We use a three-layer net with hidden dimensions 50-10, which has more parameters than the architecture with restricted approximability. Results are plotted in Figure 5. We find that the generators also converge well in the KL divergence, but the correlation is slightly weaker than the setting with restricted approximability (correlation still presents along each training run but weaker across different runs). This suggests that vanilla discriminator structures might be practically quite satisfying for getting a good generator, though specific designs may help improve the quality of the distance $W_\mathcal{F}$.

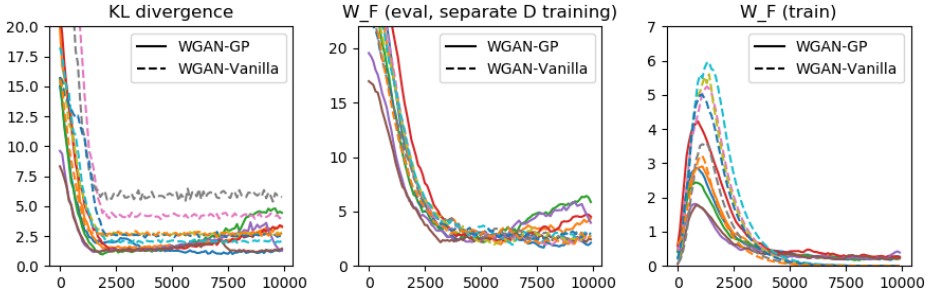

Figure 5: Learning an invertible neural net generator on synthetic data with vanilla fully-connected discriminator nets. The x-axis in all the graphs indicates the number of steps. The left-most figure shows the KL-divergence between the true distribution $p$ and learned distribution $q$ at different steps of training, the middle the estimated IPM (evaluation) between $p$ and $q$, and the right one the training loss. We see that the estimated IPM in evaluation correlates well with the KL-divergence. Moving average is applied to all curves.

#### G.4.2 PERTURBED GENERATORS

Correlation between KL and neural net IPM is computed with vanilla fully-connected discriminators and plotted in Figure 6. The correlation (0.7489) is roughly the same as for discriminators with restricted approximability (0.7315).

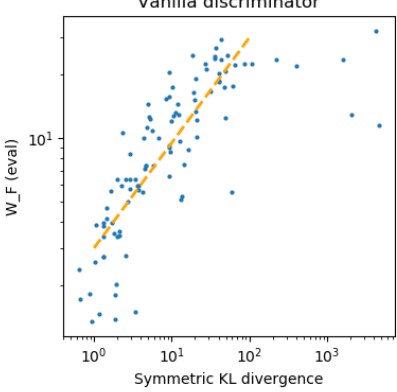

Figure 6: Scatter plot of KL divergence and neural net IPM (with vanilla discriminators) on perturbed generator pairs. Correlation between $\log(D_{\mathrm{kl}}(p\|q) + D_{\mathrm{kl}}(q\|p))$ and $\log W_\mathcal{F}$ is 0.7489. Dashed line is $W_\mathcal{F}(p, q) = 3\sqrt{D_{\mathrm{kl}}(p\|q) + D_{\mathrm{kl}}(q\|p)}$.

