# OpenReview forum: "Approximability of Discriminators Implies Diversity in GANs"
_ICLR.cc/2019/Conference_

### Official Review · AnonReviewer3 · 2018-10-31
**Good paper**

**Rating:** 7
**Confidence:** 3

**Review:**

This paper analyzes that the Integral Probability Metric (IPM) can be a good approximation of Wasserstein distance under some mild assumptions. They first showed two theorems based on simple cases (Gaussian Distribution and Exponential Families). Then, they proved that, for an invertible generator, a special designed neural network can approximate Wasserstein distance with IPM. The main contribution is that, for a stable generator (i.e., invertible generator), a discriminator can reversely “re-visit” inner status of the generator, then use this information to make a decision.

In the appendix, several numerical examples are presented to support their theoretical bound.

Q: Assumption 1, \sigma(t) is twice differentiable. However, Leaky ReLU is not twice differentiable at t=0. Do I misunderstand some part?

Q: The invertible generator assumption is not held in practice. Is that possible to extend the theorem to this case, even with a shallow network (e.g. 2 layers)?

Q: The numerical examples are all based on synthetic data. Did you have any results based on the real dataset?

---

> ### Author Response · Authors · 2018-11-27
> **Response**
>
> Thank you for the valuable feedback! We respond to the specific questions in the following.
>
> --- “Theory on NN generators requires \sigma twice differentiable”
> Though the leaky ReLU is not twice differentiable (not even differentiable everywhere), we gave the example that a smoothed version of Leaky ReLU does satisfy the assumption after we present Assumption 1. In our experiments, we did use the original Leaky ReLU activation, which works well.
>
> --- “Invertible generator does not hold in practice”
> In Section 4.2, we have results on injective generators, which is more general than invertibility. Other more involved cases are left for future work.
>
> --- “Real vs. synthetic experiment”
> We did our experiments on synthetic datasets because in order to test the theory, we need to evaluate at least one “true” distance metric (such as KL or Wasserstein) between the generated distribution and the true distribution. The KL divergence can only be computed for invertible generators which, as the reviewer suggested, does not hold for real data. Wasserstein cannot be estimated for high-dimensional real data *from samples* because accurate estimation of Wasserstein requires an exponential (in dimension) number of samples.
>
> Therefore, we designed synthetic experiments so that either we know that the invertible generator can fit the data and we can compute the KL divergence (in Appendix G), or we use 2d synthetic data (in Section 5.1) for which we can compute the Wasserstein distance.
>
> That being said, it would be a good future direction to see how our theory (or its implication) can be tested on real data. For example, to see whether the “evaluation IPM” (see e.g. Section G.1)  is well-correlated with other commonly used metrics in GANs, such as the inception score.

---

### Official Review · AnonReviewer1 · 2018-11-01
**Proposes the notion of restricted approximability, and provides a sample complexity bound, polynomial in the dimension, for GANs. Whether it proposes use of properly-designed discriminator architecture in GAN learning is not clear enough.**

**Rating:** 7
**Confidence:** 3

**Review:**


[pros]
This paper proposes the notion of restricted approximability, and provides a sample complexity bound, polynomial in the dimension, for GANs.
The proposal is especially useful in investigating possible cause of the lack of diversity in GANs.

[cons]
Whether it proposes use of properly-designed discriminator architecture in GAN learning is not clear enough.
The claimed ability of the proposed method to avoid mode collapse is not directly addressed in the experiments presented in the appendices.

[quality]
The contents of Section 3 may be useful as case studies but are not used in the following sections on neural network generators. It would thus be better to include experimental results into the main part of the paper rather than the current contents in Section 3.

[clarity]
In most parts of this paper, the authors seem to propose designing a proper discriminator architecture according to the generator class, and the discriminator architecture is to be used in GAN learning. It seems, however, that a "properly-designed discriminator architecture" is not used at all in the experiments in Appendix F. A comparison between a "properly-designed discriminator architecture" and a "vanilla fully-connected distriminator" is found in Appendix G.4, where the advantage of the former seems marginal. The authors also seem to use the proposal not to improve GAN learning but rather as a tool for evaluation, in order to see whether the lack of diversity in GANs comes either from failure of properly evaluating the Wasserstein distance or from insufficient optimization in learning. These two distinct subjects are discussed in a mixed way, which reduces clarity of the presentation.
In the experiments in Appendix G, it is claimed that a discriminator with the architecture specified in Lemma 4.1 is used in GAN learning, but either weight clamping or gradient penalty is used as well. It is unclear how the specifications in Lemma 4.1 for the parameter $\phi$ are combined with weight clamping or gradient penalty.
Some statements include forward reference, which obscure readability. For example, in the last paragraph of Section 1.1 "the statistical properties of GANs" are mentioned without an explicit statement as to what they mean, which are given later in page 3, lines 6-12. As another example, in the third paragraph of Section 1.3 the authors start discussing the KL-divergence, but at this point it is not evident at all why they do it. It is not until Section 4.1 that the reader can understand the reason by observing that the main theorem (Theorem 4.2) is proved by making use of KL-divergence.

[originality]
The idea of introducing the notion of restricted approximability and discussing a sample complexity bound, polynomial in the dimension, for GANs are considered original.

[significance]
The whole arguments in this paper are based on the assumption that both $p$ and $q$ are in the class $\mathcal{G}$. In the context of GAN learning, it poses no problem for the generator since we explicitly parameterize it, for example using a neural network, but in practice there is no guarantee that the target distribution also belongs to the same class, and this point would affect significance of the proposal. One may argue that when one employs a certain neural network architecture for the generator one expects that the target distribution is well expressed by a network with the prescribed architecture. But the question as to what will happen when the target distribution does not belong to the class $\mathcal{G}$ remains. In any case, no discussion is presented in this paper as for this question.

[minor points]
Page 3, line 45: for low-dimensional (dimensions -> distributions)

Page 4, line 8: Remove the parentheses enclosing Lopez-Paz & Oquab, 2016.

Page 4, lines 20-21: Duplicate parentheses.

Page 4, line 7: the true and estimated distribution(s) exist.

Page 5, line 33: the lower and upper bound(s) differ

Page 7, line 9: What do "some assumptions" refer to?

Page 8, line 44: The(re) exists a discriminator class

Page 19, line 1: there exi(s)ts a coupling

---

> ### Author Response · Authors · 2018-11-27
> **Response**
>
> Thank you for the thoughtful comments! We have revised the paper to address several questions raised in the comments (which we detail below). We now respond to the specific concerns.
>
> --- The reviewer suggested swapping the contents of Section 3 with experiments in the Appendix.
>
> We have migrated our 2d synthetic data experiment (previously Appendix F) into the main text as Section 5.1. The setups and results of our other experiment are still reported at the beginning of Section 5 and detailed in Appendix G.
>
> --- “Relation between proposed discriminator design and experiments”
>
> The goal of our specific discriminator designs is indeed not to make them better than vanilla ones, but rather to show that the restricted approximability can be achieved in principle. Our experiments verify this and show that choosing either our specific design or vanilla discriminators of reasonable capacity will yield good statistical property (IPM correlates well with KL / W_1). We note that indeed our theory suggests that the discriminator should have 2 more layers than the generator.
>
> We have added a few paragraphs in Section 5 to clarify the purpose of our experiments, and their relationship with our theory.
>
> --- “Relationship between designed discriminator F and weight clamping / gradient penalty”
> Some sort of weight clamping / constraints for F is needed in any case, since F has to be a bounded set so that the IPM is bounded. We imposed operator norm constraints on the weight matrices that is particularly compatible with the theory, whereas typical weight clamping is in a stronger entry-wise fashion (e.g. in Arjovsky et al. ‘17).
>
> The gradient penalty (GP) is indeed introduced to help the optimization in the experiments, and when we add GP we still keep the constraint on the operator norm of the weight matrices. Therefore, in our experiments in Appendix G, GP does not change the search space for the discriminators but is rather an alternative to SGD for finding a discriminator in this space.
>
> --- “What happens when the target distribution is not in G”
> Our theory builds on the assumption that p is in G (the realizable assumption), but the extension to the non-realizable case can be done straightforwardly. For example, if F can approximate log p - log q for all q in G and the data distribution p, then all the theory in the paper still holds. Indeed, our Lemma 4.3 is stated in this general form without assuming p is in G. We have revised Section 1.2 to allow such generality and mentioned that results in the non-realizable case can be established, for example, through applying Lemma 4.3.

---

### Official Review · AnonReviewer2 · 2018-11-02
**Interesting theoretical work on establishing sample complexity bounds for learning certain distributions using GANS**

**Rating:** 8
**Confidence:** 2

**Review:**

This paper explores how discriminators can be designed against certain generator classes to reduce mode collapse. The strength of the paper is on establishing the sample complexity bounds for learning such distributions to show why they can be effectively learned. The work is important in understanding the behaviour of GANs. The work is original and significant. A few comments that need to be addressed are listed as below:

1. I found the paper is a bit hard to follow in the beginning, due to its structure. In Section 1, it first gives introduction and then talks about the novelty of the paper; it then shows more background work followed by more introduction of the proposed work; after that, Section 1.4 talks more related work. It makes reading confusing in the beginning.

2. The authors wrote that "In practice, parametric families of functions F such as multi-layer neural networks are used for approximating Lipschitz functions, so that we can empirically optimize this objective eq. (2) via gradient-based algorithms as long as distributions in the family G have parameterized samplers. (See Section 2 for more details.)" I am not sure how Section 2 gives more details.

3. There are some typos and the references are not very carefully edited. For example, in Theorem 4.5, "the exists a ..." -> "there exists a ..."; in reference, gan -> GAN.

---

> ### Author Response · Authors · 2018-11-27
> **Response**
>
> Thank you for the positive feedback. We have revised our paper to address the specific questions. We respond to the specific comments in the following.
>
> 1. “Structure of Section 1”
> We have added a few pointers at the beginning of the paper (before Section 1.1) to clarify the structure of the entire introduction section.
>
> 2. We have removed the “see Section 2 for more details” for clarity.
>
> 3. We have edited the reference, as well as the typo in Theorem 4.5.

---

### Public Comment · ~Hello_Kitty2 · 2018-10-03
**numerical results**

i read this paper a couple months ago. It is probably one of the best papers i have read in the past one year. It has extended the results of GAN in the LQG setting to a more general setup. The paper has provided insightful theoretical understanding of the dynamics of GAN.

i have a couple questions:
(1) if the discriminator is not properly constrained, the generalization may be slow. could the authors provide some numerical results showing the negative effects of generalization when the discriminator is not properly constrained?
(2) many papers propose different regularization/normalization techniques (e.g., spectral normalization, etc.), which give very good performance. is it because such techniques somehow constrain the neural networks and thus give better generalization properties?

---

> ### Author Response · Authors · 2018-10-10
> **Thanks & Response**
>
> Thank you for the comments, and we’re glad that you liked our paper!
>
> Regarding the numerical results:
>
> (1) Empirically, the discriminator has to be constrained in order to even track a meaningful distance between distributions (see, e.g. the Wasserstein GAN paper by Arjovsky et al.) Our paper shows that, from a statistical point of view, a properly restricted discriminator and generator class in Wasserstein GANs can provably learn the data distribution with a polynomial sample complexity bound.
>
> Our particular construction for injective neural nets requires :
> a) the discriminators to have two more layers than the generator (with a particular structure) to ensure distinguishability;
> b) the number of samples to be larger than the discriminator complexity (but still polynomial in dimension) to ensure generalization.
>
> (2) Normalization / regularization techniques such as the spectral normalization very likely help the generalization through reducing the capacity of the discriminator class and potentially also help the optimization. Note however that merely having a nice discriminator class (potentially constrained/regularized) cannot guarantee generalization (Arora et al., 2017a), because the generator may output a mode-dropped distribution that fools all discriminators in the designed class.
>
> Indeed, the key message of our paper is that when the generator, as well as the true distribution, belong to certain structured families, and discriminators “collaborate” well with the generator class (i.e. have restricted approximability), the distribution is learnable with polynomial number of samples. We give an instantiation of the framework for injective neural nets, with a discriminator class that works well empirically on synthetic data. However, generally, the “optimal” choice of discriminators for commonly used generator classes is still an open question and interesting direction for future work.

---

### Author Response · Authors · 2018-11-27
**Update: paper revised**

We have made a revision to our paper according to the reviewers' comments. Major changes are the following:

--- We have migrated one set of our experiments (previously Appendix F) into the main text (Section 5.1).

--- The concept of restricted approximability is now defined in more generality without assuming the data distribution is realizable by the generator class (Section 1.2).

---

### Meta-Review · Area_Chair1 · 2018-12-08
**Interesting theoretical work proving sample complexity bounds for GAN training**

**Confidence:** 4
**Recommendation:** Accept (Poster)

**Metareview:**

The paper presents an interesting theoretical analysis by deriving polynomial sample complexity bounds for the training of GANs that depend on the approximator properties of the discriminator.
Even if it is not clear if the theory will help to pick suitable discriminators in practice, it provides
new and interesting theoretical insights on the properties of GAN training.